# OneForecast: A Universal Framework for Global and Regional Weather Forecasting

**Yuan Gao** [* 1]   **Hao Wu** [* 1 2 3]   **Ruiqi Shu** [* 1]   **Huanshuo Dong** [3]   **Fan Xu** [3]   **Rui Ray Chen** [4]   **Yibo Yan** [5 6]
**Qingsong Wen** [7]   **Xuming Hu** [5 6]   **Kun Wang** [8]   **Jiahao Wu** [9]   **Qing Li** [9]   **Hui Xiong** [5 6]   **Xiaomeng Huang** [1]

## Abstract

Accurate weather forecasts are important for disaster prevention, agricultural planning, etc. Traditional numerical weather prediction (NWP) methods offer physically interpretable high-accuracy predictions but are computationally expensive and fail to fully leverage rapidly growing historical data. In recent years, deep learning models have made significant progress in weather forecasting, but challenges remain, such as balancing global and regional high-resolution forecasts, excessive smoothing in extreme event predictions, and insufficient dynamic system modeling. To address these issues, this paper proposes a global-regional nested weather forecasting framework (OneForecast) based on graph neural networks. By combining a dynamic system perspective with multi-grid theory, we construct a multi-scale graph structure and densify the target region to capture local high-frequency features. We introduce an adaptive messaging mechanism, using dynamic gating units to deeply integrate node and edge features for more accurate extreme event forecasting. For high-resolution regional forecasts, we propose a neural nested grid method to mitigate boundary information loss. Experimental results show that OneForecast performs excellently across global to regional scales and short-term to long-term forecasts, especially in extreme event predictions. Codes link: https://github.com/YuanGao-YG/OneForecast.

---

[*]Equal contribution [1]Department of Earth System Science, Ministry of Education Key Laboratory for Earth System Modeling, Institute for Global Change Studies, Tsinghua University [2]TEG, Tencent [3]Department and Computer and Science, University of Science and Technology of China [4]Institute for Interdisciplinary Information Sciences, Tsinghua University [5]Department of Computer Science and Engineering, The HongKong University of Science and Technology [6]AI Thrust, The Hong Kong University of Science and Technology (Guangzhou) [7]Squirrel Ai Learning [8]School of Computer Science and Engineering, Nanyang Technological University [9]Department of Computing, The Hong Kong Polytechnic University. Correspondence to: Xiaomeng Huang <hxm@tsinghua.edu.cn>.

*Proceedings of the 42nd International Conference on Machine Learning*, Vancouver, Canada. PMLR 267, 2025. Copyright 2025 by the author(s).

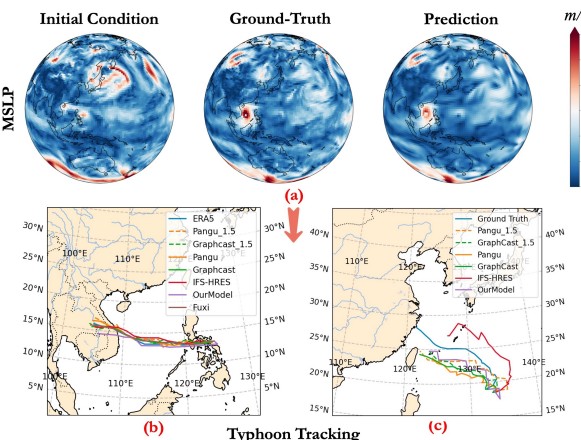

*Figure 1.* Forecast results of extreme typhoons. (a) OneForecast's predicted wind speed for Typhoon Molva (2020) at 850 hPa pressure level with a 60-hour lead time. (b)–(c) the predicted cyclone tracks of Typhoon Yagi (2018) and Typhoon Molva (2020) using different models

## 1. Introduction

Accurate weather forecasting is crucial for disaster prevention, optimizing agriculture and energy planning, and ensuring water resource management (Chen et al., 2023a; Pathak et al., 2022; Ahmed et al., 2021). Traditional Numerical Weather Prediction (NWP) methods (Bauer et al., 2015) rely on the numerical solution of atmospheric dynamic equations (Achatz et al., 2023; Buzzicotti et al., 2023), ensuring consistency across spatiotemporal scales from a physical perspective. However, with the growing volume of observational and historical data, and the increasing demand for high-resolution and long-term forecasts, NWP methods often struggle with computational costs and fail to fully leverage the potential value of vast data.

Recent developments in deep learning (DL) models offer new perspectives for weather forecasting. Early spatiotemporal prediction algorithms (Wu et al., 2024a), such as ConvLSTM (Shi et al., 2015) and PredRNN (Wang et al., 2022), focus on regional precipitation. Recent large-scale scientific computing models, like Pangu-weather (Bi et al., 2023), GraphCast (Lam et al., 2023), and NowcastNet (Zhang et al., 2023), achieve significant results in medium- and short-term forecasts and show high potential in extreme event prediction (e.g., precipitation and Typhoon track prediction) (Chen et al., 2024; Espeholt et al., 2022; Gong et al., 2024; Wu et al., 2024b). However, pure AI

methods still face several core challenges:

❶ *Global and regional high-resolution forecasts are hard to balance.* Regional predictions often lack boundary information, making it difficult to effectively nest global data. ❷ *Extreme events and long-term forecasts suffer from over-smoothing.* They fail to capture high-frequency disturbances, leading to reduced forecast accuracy. ❸ *Lack of dynamic system modeling capability.* This is especially true for capturing complex interactions between nodes at multiple scales and learning high-frequency node-edge features.

To address these challenges, we propose OneForecast, a global-regional nested weather forecasting framework based on Graph Neural Networks (GNNs). Inspired by heuristic learning from numerical methods, we construct a multi-scale graph structure based on dynamical systems and multi-grid theory (He & Xu, 2019; He et al.), refining it for the target region to capture local high-frequency features with greater detail. Additionally, to solve issues like over-smoothing in extreme events and long-term forecasts, which hinder the capture of high-frequency disturbances, we introduce an adaptive information propagation mechanism. This mechanism deepens the integration of node and edge features through dynamic gating units. Finally, for regional high-resolution forecasting, we adopt a nested grid strategy (Phillips & Shukla, 1973) that inherits large-scale background information from the global scale, significantly alleviating the boundary information loss. *Through this integrated framework, we aim to effectively capture high-frequency features and extreme events across global to regional scales, as well as from short-term to long-term forecasts.*

The method most similar to ours is Graph-EFM (Oskarsson et al., 2024). It also uses a hierarchical graph neural network for global and regional weather modeling. However, for high-resolution regional forecasts, it treats global low-resolution data as non-trainable forcing conditions, making it unable to adaptively couple multi-scale information based on actual needs. And it doesn't treats the forecasts of the global model in the region as forcing, which unable to fully utilize the information of the global model. In contrast, our neural nested grid method applies trainable local refinement to the target region in the network structure. It also updates boundary and background information with global model future forcing dynamically through end-to-end training during global-regional coupling. This design better captures the interaction between large-scale global backgrounds and high-frequency regional details. Our experiments (Sec 3.3) show that our method achieves greater stability and accuracy in long-term rolling inference.

The contribution of this paper can be summarized as follows: (1) *Global-Regional Unified Forecasting Framework.* We propose a Graph Neural Network method that supports both global scale and regional high-resolution forecasting,

achieving high-accuracy results for multi-scale and multi-time frame weather forecasts within the same framework. (2) *Adaptive Information Propagation Mechanism.* Through Dynamic Gating Units and graph attention modules, we deeply integrate node and edge features, more accurately capturing extreme events and other high-frequency disturbance signals within the multi-scale graph structure. As shown in Figure 1, OneForecast delivers better performance in tracking extreme events like typhoons. (3) *Nested Grid and Long-Term Forecasting.* By using a nested grid to merge global and regional information, we overcome the boundary loss issue in regional forecasting. This method effectively mitigates the loss of details caused by over-smoothing in long-term forecasts.

## 2. Method

**Problem Definition** In this study, we model weather forecasting as an autoregressive problem (Lam et al., 2023). At each time step $t$, we use the meteorological state comprising surface variables $\mathbf{X}_t$ and pressure level variables $\mathbf{P}_t$ to forecast the state at the next time step. We concatenate the surface and pressure level variables along the channel dimension to form the combined input: $\mathbf{Z}_t = [\mathbf{X}_t, \mathbf{P}_t] \in \mathbb{R}^{N \times d}$, where $N = H \times W$ represents the number of grid locations (nodes), and $d = d_x + d_p$ is the total number of variables. Here, $d_x$ and $d_p$ are the numbers of surface level and pressure level variables, respectively. In our setup, the initial input contains 69 variables: 4 surface level variables and 65 pressure level variables. Our model aims to forecast the combined variables at the next time step $\hat{\mathbf{Z}}_{t+1}$ using the current input $\mathbf{Z}_t$, capturing the spatiotemporal evolution of the atmosphere: $\hat{\mathbf{Z}}_{t+1} = \text{Model}(\mathbf{Z}_t; \Theta)$, where $\Theta$ denotes the model parameters. The training objective is to minimize the relative mean squared error (MSE) between the forecasts and the true values across all time steps: $\min_{\Theta} \frac{1}{T} \sum_{t=0}^{T-1} \frac{\|\hat{\mathbf{z}}_{t+1} - \mathbf{z}_{t+1}\|_2^2}{\|\mathbf{Z}_{t+1}\|_2^2}$. During inference, we adopt a rollout strategy to forecast longer sequences. Starting from the initial state $\mathbf{Z}_0$, the model recursively uses its previous forecasts as the next input: $\hat{\mathbf{Z}}_{t+1} = \text{Model}(\hat{\mathbf{Z}}_t; \Theta), \quad t = 0, 1, 2, \ldots, T-1$. This strategy allows the model to generate extended weather forecasts using its own forecasts.

### 2.1. Earth-specific Region Refined Graph Encoder

In the encoder of OneForecast, inspired by (Fortunato et al., 2022) (Lam et al., 2023), we introduce an Earth-specific Region Refined Multi-scale Graph to improve the interaction of node features in complex dynamical systems. Inspired by the idea of multigrid methods (He & Xu, 2019), we construct a multi-level Graph Neural Network architecture that includes grids of multiple granularities. Each grid has the same number of nodes but different grid densities, thereby

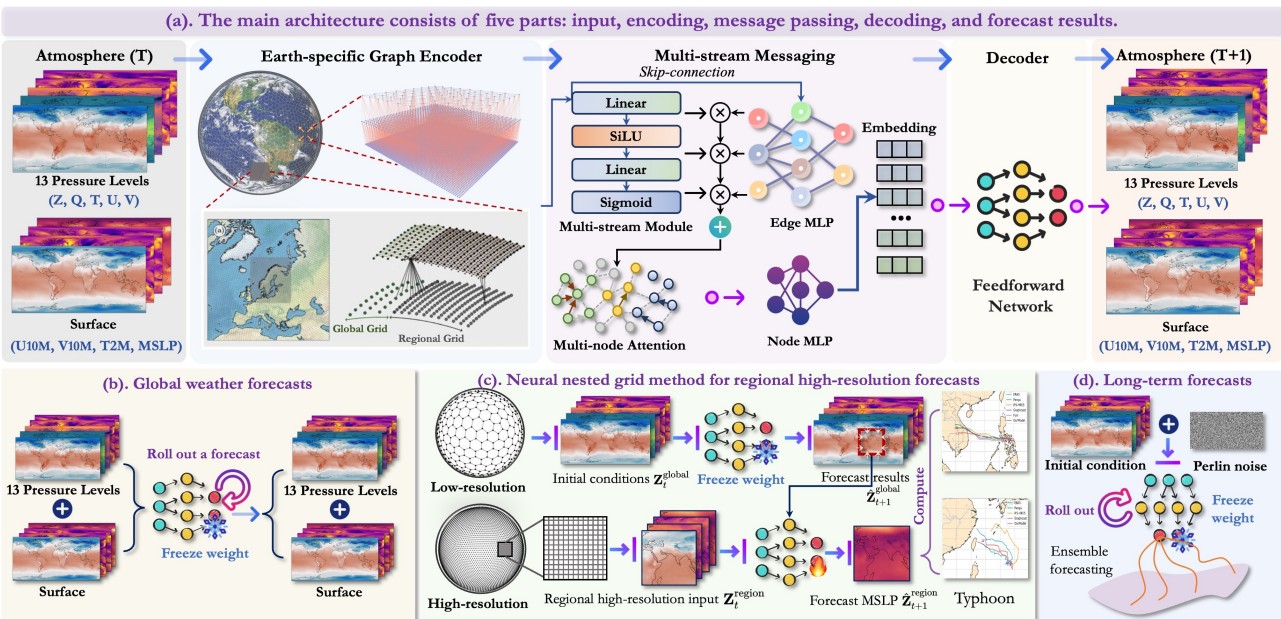

*Figure 2.* Overview of Our OneForecast. **(a)** The overall architecture includes input variables, an encoder, a message passing module, a decoder, and visualization of forecast variables; **(b)** The global forecasts module uses rollout technology to generate future forecasts; **(c)** The neural nested grid method specializes in regional high-resolution weather forecasts tasks; and **(d)** The ensemble forecasting module generates long-term forecast results.

capturing spatial features at different scales. Specifically, we define the multi-scale graph structure as:

$$\mathcal{G} = \left( \mathcal{V}^G, \mathcal{V}, \mathcal{E}^{(1)}, \mathcal{E}^{(2)}, \ldots, \mathcal{E}^{(L)}, \mathcal{E}^{(R)}, \mathcal{E}^{\text{G2M}}, \mathcal{E}^{\text{M2G}} \right), \quad (1)$$

where $\mathcal{V}^G$ represents the set of lat-lon grid nodes, with a total of $N = H \times W$ nodes; $\mathcal{V}$ represents the mesh nodes. $\mathcal{E}^{(l)}$ denotes the edge set at the $l$-th scale, corresponding to grids of different granularities, where $l = 1, 2, \ldots, L$, and $\mathcal{E}^{(R)}$ represents regional refined edges. $\mathcal{E}^{\text{G2M}}$ and $\mathcal{E}^{\text{M2G}}$ are the unidirectional edges that connect lat-lon grid nodes and mesh nodes. All scales share the same set of nodes $\mathcal{V}$. More details can be found in Appendix E.

In the encoder, we first map the input meteorological state $\mathbf{Z}_t \in \mathbb{R}^{N \times d}$ to the initial node feature representation:

$$\mathbf{h}_i^{(0)} = \phi(\mathbf{Z}_{t,i}), \quad i = 1, 2, \ldots, N, \quad (2)$$

where $\phi(\cdot)$ is the feature mapping function, and $\mathbf{Z}_{t,i}$ is the input feature at node $i$. Next, we iteratively update the node features on the multi-scale graph structure. At iteration $k$, the feature update formula for node $i$ is:

$$\mathbf{h}_i^{(k)} = \sigma \left( \sum_{l=1}^{L} \sum_{j \in \mathcal{N}_i^{(l)}} \mathbf{W}^{(l)} \mathbf{h}_j^{(k-1)} + \mathbf{b}^{(l)} \right), \quad (3)$$

where $\mathcal{N}_i^{(l)}$ is the set of nodes adjacent to node $i$ at the $l$-th scale, $\mathbf{W}^{(l)}$ is the weight matrix at the $l$-th scale, $\mathbf{b}$

is the bias term, and $\sigma(\cdot)$ is the activation function. To enhance the forecasting accuracy in specific regions, we introduce a region-refined grid on the finest global grid. For nodes within the target region, we add denser edge connections to capture local high-frequency features. In this way, the update of node features not only considers global multi-scale information but also incorporates region-specific fine-grained information.

### 2.2. Multi-stream Messaging

To address the issue of information transmission between nodes in complex dynamic systems, we propose a module called *Multi-stream Messaging* (MSM). This module consists of an adaptive messaging mechanism, including a dynamic multi-head gated edge update module and a multi-head node attention mechanism. And OneForecast includes 16 MSMs for messaging.

**Dynamic Multi-head Gated Edge Update Module.** Unlike traditional message passing methods based on MLPs, we introduce dynamic gating and multi-head mechanisms to control the information flow more precisely. For each edge, we concatenate its own features with those of the source node and the target node:

$$\mathbf{c}_i = \text{Concat}\left( \mathbf{e}_i, \mathbf{h}_{s(i)}, \mathbf{h}_{d(i)} \right) \in \mathbb{R}^{D_e + 2D_h}, \quad (4)$$

where $\mathbf{e}_i$ is the feature of edge $i$, $\mathbf{h}_{s(i)}$ and $\mathbf{h}_{d(i)}$ are the features of the source and target nodes of edge $i$, $D_e$ is the edge feature dimension, and $D_h$ is the node feature

dimension. Next, we generate gating vectors through a two-layer MLP to regulate the information flow. Specifically, the gating vector is divided into three parts: edge feature update gate $g_e$, source node feature gate $g_s$, and destination node feature gate $g_d$. First, we perform the first layer linear transformation and activation:

$$\mathbf{z}_i = \text{SiLU}\left(\mathbf{W}_1\mathbf{c}_i + \mathbf{b}_1\right), \tag{5}$$

where $\mathbf{W}_1 \in \mathbb{R}^{h \times (D_e + 2D_h)}$ is the weight matrix of the first layer, and $\mathbf{b}_1 \in \mathbb{R}^h$ is the bias term. Then, we perform the second layer linear transformation and Sigmoid activation:

$$\mathbf{g}_i = \sigma\left(\mathbf{W}_2\mathbf{z}_i + \mathbf{b}_2\right) \in \mathbb{R}^{3HD}, \tag{6}$$

where, $\mathbf{W}_2 \in \mathbb{R}^{3HD \times h}$ is the weight matrix of the linear transformation of the second layer, and $D$ is the feature dimension for each gate. For each head $h$, the gating values $\mathbf{g}_i^{(h,e)}$, $\mathbf{g}_i^{(h,s)}$, and $\mathbf{g}_i^{(h,d)}$ are vectors of dimension $D$, corresponding to the edge feature gate, source node feature gate, and destination node feature gate, respectively. Subsequently, we use Edge Sum MLP (ESMLP) (Pfaff et al., 2020) to perform linear transformation and nonlinear activation on the edge features to generate the updated edge features:

$$\mathbf{e}_i' = \text{ESMLP}_e\left(\mathbf{e}_i, \mathbf{h}_{s(i)}, \mathbf{h}_{d(i)}\right) \in \mathbb{R}^{D_e'}, \tag{7}$$

where $D_e'$ is the dimension of the updated edge features. Finally, we combine the gating vectors and the updated edge features to generate the final updated edge features through weighted averaging and residual connections:

$$\begin{aligned}
\mathbf{e}_i^{\text{new}} = \frac{1}{3}\sum_{h=1}^{H}\Bigg(&\mathbf{g}_i^{(h,e)} \odot \mathbf{e}_i' + \mathbf{g}_i^{(h,s)} \odot \mathbf{h}_{s(i)} \\
&+ \mathbf{g}_i^{(h,d)} \odot \mathbf{h}_{d(i)}\Bigg) + \mathbf{e}_i,
\end{aligned} \tag{8}$$

where $\odot$ denotes element-wise multiplication.

**Multi-head Node Attention Mechanism.** Compared to traditional message passing mechanisms, multi-head attention mechanisms can more precisely capture complex dependencies between nodes and dynamically adjust the way information is aggregated through attention weights. For each edge $e_i = (j \to k)$, we use a MLP to calculate the attention score:

$$\mathbf{a}_i = \text{MLP}_a\left(\mathbf{e}_i^{\text{new}}\right) \in \mathbb{R}^H. \tag{9}$$

Then, we normalize the attention scores:

$$\alpha_i^{(h)} = \frac{\exp\left(\mathbf{a}_i^{(h)}\right)}{\sum_{e_j \in \mathcal{E}(k)}\exp\left(\mathbf{a}_j^{(h)}\right)}, \quad \forall h = 1, 2, \ldots, H, \tag{10}$$

where $\mathcal{E}(k)$ denotes the set of all incoming edges to node $k$, and $\alpha_i^{(h)}$ is the attention weight of the $h$-th attention head for edge $e_i$. Next, we perform weighted aggregation of the edge features. For each node $k$, based on the attention weights, we compute the weighted sum of the features of all edges incoming to node $k$, generating the aggregated feature for each head:

$$\mathbf{m}_k^{(h)} = \sum_{e_i \in \mathcal{E}(k)} \alpha_i^{(h)} \cdot \mathbf{e}_i^{\text{new}} \in \mathbb{R}^{D_e'}. \tag{11}$$

Then, we flatten and concatenate the aggregated features from all heads:

$$\mathbf{M}_k = \text{Flatten}\left[\mathbf{m}_k^{(1)}, \mathbf{m}_k^{(2)}, \ldots, \mathbf{m}_k^{(H)}\right] \in \mathbb{R}^{D_e' \cdot H}. \tag{12}$$

Finally, we concatenate the aggregated edge features with the original node features and, through a MLP, perform a nonlinear transformation to generate the updated node features:

$$\mathbf{h}_k^{\text{new}} = \text{MLP}_n\left(\text{Concat}\left(\mathbf{M}_k, \mathbf{h}_k\right)\right) + \mathbf{h}_k. \tag{13}$$

In summary, in each iteration, we use the multi-stream messaging module to update the node features. Specifically, the node feature update formula is:

$$\mathbf{h}_i^{(k)} = \sigma\left(\sum_{l=1}^{L}\text{MSM}\left(\mathbf{h}_i^{(k-1)}, \mathcal{E}^{(l)}\right) + \mathbf{b}\right), \tag{14}$$

where MSM represents the aforementioned multi-stream messaging operation, $\mathcal{E}^{(l)}$ is the set of edges at the $l$-th scale, and $\sigma(\cdot)$ is the activation function. In the region-refined graph structure, for nodes within the target region, we additionally consider the set of edges within the region $\mathcal{E}^{\text{region}}$ to capture finer local information.

**Theoretical Analysis.** From a theoretical perspective, we explain why our method helps capture high-frequency information. This enhances long-term prediction ability and improves the ability to detect extreme events.

**Theorem 2.1.** *High-pass Filtering Property of Multi-stream Messaging. Considering the improved multi-stream message passing mechanism, suppose the graph signal $\boldsymbol{f} \in \mathbb{R}^N$ has a spectrum $\hat{\boldsymbol{f}} = \boldsymbol{U}^\top \boldsymbol{f}$ under the graph Fourier basis $\boldsymbol{U} = [\boldsymbol{u}_1, ..., \boldsymbol{u}_N]$, where $\boldsymbol{L} = \boldsymbol{U}\boldsymbol{\Lambda}\boldsymbol{U}^\top$ is the normalized graph Laplacian matrix and $\boldsymbol{\Lambda} = \text{diag}(\lambda_1, ..., \lambda_N)$ is its eigenvalue diagonal matrix ($0 \leq \lambda_1 \leq ... \leq \lambda_N \leq 2$). Define the frequency response function of the message passing operator as $\rho : \lambda \mapsto \mathbb{R}$. If the dynamic gating weights satisfy:*

$$g_i^{(h,e)}, g_i^{(h,s)}, g_i^{(h,d)} \propto |\lambda_i - 1| + \epsilon \quad (\epsilon > 0) \tag{15}$$

*then there exist constants $\alpha > 0$ and $\kappa > 0$ such that the frequency response of the operator satisfies:*

$$\rho(\lambda_i) \geq \alpha|\lambda_i - 1| \quad and \quad \rho(\lambda_i) \geq \kappa\lambda_i \qquad (16)$$

*that is, the operator is a strictly high-pass filter.*

The proof of Theorem A.1 can be found in Appendix A.

### 2.3. Decoding and Optimization

The decoder's goal is to decode the latent information back to meteorological variables on the latitude-longitude grid. We obtain the updated feature representation $\mathbf{h}_i$ for each node. For each node $i$, the decoder applies the mapping function:

$$\hat{\mathbf{Z}}_{t+1,i} = \psi(\mathbf{h}_i), \qquad (17)$$

where $\psi(\cdot)$ is an MLP that converts the latent node features into the predicted variables $\hat{\mathbf{Z}}_{t+1,i}$ for the next time step.

We use relative $L_2$ loss function for model training. The loss function is defined as:

$$\mathcal{L} = \frac{1}{KHW} \sum_{k=1}^{K} \sum_{i=1}^{H} \sum_{j=1}^{W} \frac{\left(\hat{x}_{i,j,k}^{t+l\delta t} - x_{i,j,k}^{t+l\delta t}\right)^2}{\left(x_{i,j,k}^{t+l\delta t}\right)^2}, \qquad (18)$$

where $\hat{x}_{i,j,k}^{t+l\delta t}$ and $x_{i,j,k}^{t+l\delta t}$ are the predicted and true values for variable (channel) $k$ at spatial location $(i, j)$ and time $t+l\delta t$; $K$ is the number of variables (channels); $H$ and $W$ are the height and width of the spatial dimensions, respectively; $\delta t$ is the time interval of single-step prediction (we use $\delta t = 6$ hours).

### 2.4. Downstream Tasks

We consider three principal downstream tasks:

**Global Weather Forecasting.** As detailed in Section 2 and illustrated in Figure 2(b), we employ a rollout approach during inference, using the trained model for multi-step extrapolation. Specifically, starting from the initial state $\mathbf{Z}_0$, the model recursively uses its previous predictions as inputs for subsequent time steps, generating a sequence of future global weather forecasts.

**Regional High-Resolution Forecasting.** To enhance the accuracy of high-resolution forecasts in specific regions, we propose a neural nested grid method, illustrated in Figure 2(c). This method combines global low-resolution future forecasts with regional high-resolution data to produce detailed forecasts for the target region. We first input the global low-resolution data at time $t$ into the pre-trained global model to obtain the global forecasts $\hat{\mathbf{Z}}_{t+1}^{g}$ at time $t + 1$. We extract $\hat{\mathbf{Z}}_{t+1}^{global1}$ from $\hat{\mathbf{Z}}_{t+1}^{g}$, which shares the same spatial range as the region, and $\hat{\mathbf{Z}}_{t+1}^{global2}$, which includes the boundary of the region (the boundary are defined as two grid points around the region). Both $\hat{\mathbf{Z}}_{t+1}^{global1}$ and $\hat{\mathbf{Z}}_{t+1}^{global2}$

are then interpolated to match the resolution of the high-resolution regional data, which are concatenate as $\hat{\mathbf{Z}}_{t+1}^{global}$ to acted as global forcing. We then combine the regional high-resolution data at time $t$ with the $\hat{\mathbf{Z}}_{t+1}^{global}$ to form the input of the regional model. The global forecasts provide the necessary boundary conditions for the regional forecasts. The regional model then produces the high-resolution forecasts for the regional state at time $t + 1$:

$$\hat{\mathbf{Z}}_{t+1}^{region} = \text{Model}_{region}\left(\text{Concat}\left(\hat{\mathbf{Z}}_{t+1}^{global}, \mathbf{Z}_t^{region}\right); \Theta_{region}\right). \qquad (19)$$

**Long-Term and Esemble Weather Forecasting.** The initial condition of the atmospheric state is uncertain, so reasonable quantification of this uncertainty is conducive to improve to forecast performance. To account for the uncertainty in the atmospheric initial state for long-term ensemble forecasting, we generate $N$ perturbed initial conditions $\mathbf{Z}_0^{(n)}$ by adding Perlin noise $\varepsilon^{(n)}$ to $\mathbf{Z}_0$ (Chen et al., 2023b). Each perturbed initial condition is input into the model, and through recursive rollout over $T$ time steps, we obtain individual forecasts $\hat{\mathbf{Z}}_{t+1}^{(n)}$. Finally, at each time step $t + 1$, we compute the ensemble mean prediction $\hat{\mathbf{Z}}_{t+1}^{ensemble} = \frac{1}{N} \sum_{n=1}^{N} \hat{\mathbf{Z}}_{t+1}^{(n)}$ by averaging the forecasts from all $N$ ensemble members. In this work, we set N=50.

## 3. Experiments

In this section, we extensively evaluate the performance of OneForecast, covering metric results, visual results, and extreme event analysis. We conduct all experiments on 128 NVIDIA A100 GPUs.

### 3.1. Benchmarks and Baselines

We conduct the experiments on the WeatherBench2 (Rasp et al., 2024) benchmark, a subset of the fifth generation of ECMWF Reanalysis Data (ERA5) (Hersbach et al., 2020). The subset we use includes years from 1959 to 2020, which is 1959-2017 for training, 2018-2019 for validating, and 2020 for testing. We use 5 pressure level variables (each with 13 pressure levels), geopotential (Z), specific humidity (Q), temperature (T), U and V components of wind speed (U and V), and 4 surface level variables 10-meter U and V components of wind (U10M and V10M), 2-meter temperature (T2M), and mean sea-level pressure (MSLP). For the global weather forecasting task, we choose the 1.5° (121 × 240 for the global data) version of WeatherBench2 as our dataset. For convenience, we just choose 120 × 240 data to train models. For the regional high-resolution weather forecasting task, we use the original 0.25° (721 × 240 for the global data) ERA5 data. More details can be found in the Appendix C. We conduct 2 types comparison, the first type is the comparison between 1-step supervised models retrained using the same framework and settings, which in-

*Table 1.* In global weather forecasting task, we compare the performance of our OneForecast with 3 baselines, which are trained in the same framework. The average results for all 69 variables of RMSE (normalized) and ACC are recorded. A small RMSE (↓) and a bigger ACC (↑) indicate better performance. The best results are in **bold**, and the second best are with underline.

| MODEL | 6-HOUR | | 1-DAY | | 4-DAY | | 7-DAY | | 10-DAY | |
|---|---|---|---|---|---|---|---|---|---|---|
| | RMSE | ACC | RMSE | ACC | RMSE | ACC | RMSE | ACC | RMSE | ACC |
| PANGU-WEATHER (BI ET AL., 2023) | 0.0826 | 0.9876 | 0.1571 | 0.9581 | 0.3380 | 0.8167 | 0.5092 | 0.5738 | 0.6215 | 0.3542 |
| GRAPHCAST (LAM ET AL., 2023) | 0.0626 | 0.9928 | 0.1304 | 0.9705 | 0.2861 | 0.8705 | 0.4597 | 0.6692 | 0.6009 | 0.4275 |
| FUXI (CHEN ET AL., 2023B) | 0.0987 | 0.9820 | 0.1708 | 0.9511 | 0.4128 | 0.7379 | 0.5972 | 0.4446 | 0.6981 | 0.2391 |
| ONEFORECAST(OURS) | **0.0549** | **0.9943** | **0.1231** | **0.9737** | **0.2732** | **0.8825** | **0.4468** | **0.6888** | **0.5918** | **0.4457** |
| ONEFORECAST(PROMOTION) | 12.24% | 0.15% | 5.54% | 0.33% | 4.50% | 1.38% | 2.81% | 2.92% | 1.51% | 4.25% |

*Figure 3.* 10-day forecast results of different models.

cludes Pangu (Bi et al., 2023), Graphcast (Lam et al., 2023), Fuxi (Chen et al., 2023b), and Ours. The second type is the comparison between the results released by WeatherBench2 (with many finetune tricks), Fengwu (Chen et al., 2023a) (results released by the author), and our finetune model.

### 3.2. Comparison with state-of-the-art methods

We utilize four metrics, RMSE, ACC, CSI, and SEDI to evaluate the forecast performance. More details can be found in F.1. Since the magnitudes of different variables vary greatly, we first normalize the 69 variables and then calculate the indicators for the 1300 initial conditions. As shown in Table 1, OneForecast achieves satisfactory performance compared with the state-of-the-art models. As shown in Figure 3, OneForecast are closer with the ground truth. We also show the forecast results of several important variables in Figure 4, which are not normalized. Our OneForecast performs better than other models. More results compared with WeatherBench2 can be found in Appendix G.4. This

improvement is primarily attributed to integrating the proposed adaptive message passing module, which enhances OneForecast's ability to model the relationships between atmospheric states across different regions of the earth that allows for the simulation of atmospheric dynamics at various spatial and temporal scales adaptively. In summary, the forecasts exhibit more consistency with the actual physical field, effectively mitigate over-smoothing, and demonstrate superior predictive performance, particularly for extreme atmospheric values.

### 3.3. Regional High Resolution Forecast

Although the training cost of low-resolution forecast models is relatively low, their prediction results lack sufficient details. However, directly training on high-resolution regional data often results in poor regional forecasts performance due to issues such as missing boundary conditions and limited data samples. Although the regional forecasts method proposed by (Oskarsson et al., 2024) improves prediction

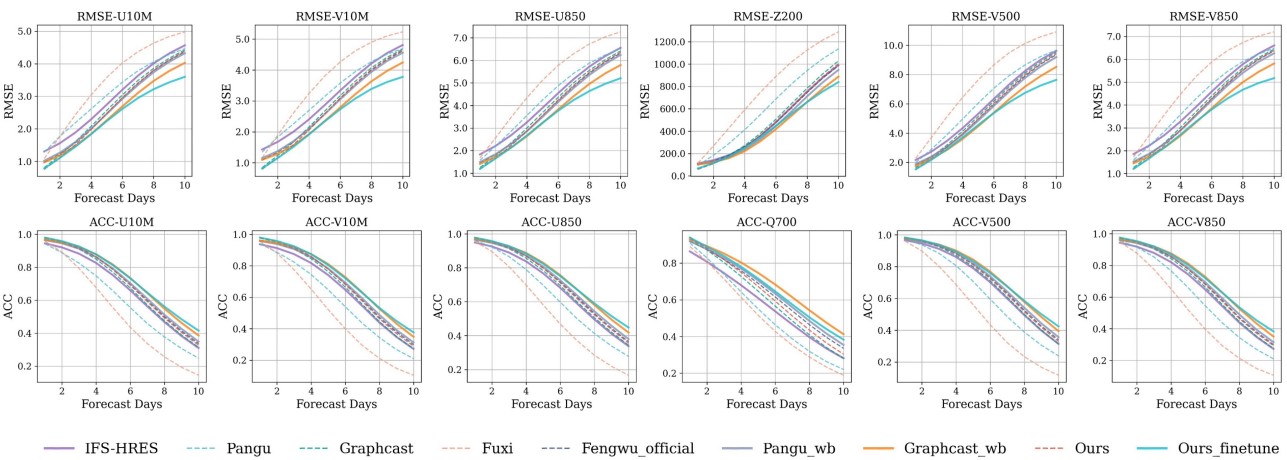

*Figure 4.* We select the latitude-weighted RMSE (lower is better) and ACC (higher is better) of several variables.

accuracy, it remains constrained by the absence of global information in high-resolution regional models. In contrast, our proposed Neural Nesting Grid method (NNG) incorporates boundary conditions and global future information, enabling more accurate high-resolution regional predictions. Furthermore, NNG makes full use of the forecast results of global models, which achieves high-resolution regional forecasts at an exceptionally low training cost. Therefore, as shown in Figure 6, we only demonstratively conduct high-resolution predictions for two regional variables without requiring training on all variables (e.g., the 69 variables used for training global models). It can be seen that the poor long-term inference results are poor when only using regional data for training. Graph-EFM takes into account boundary conditions and the effect is improved. And our proposed NNG not only takes into account regional boundary conditions, but also makes full use of the future forecast information of the global model, which achieves stable long-term forecast performance.

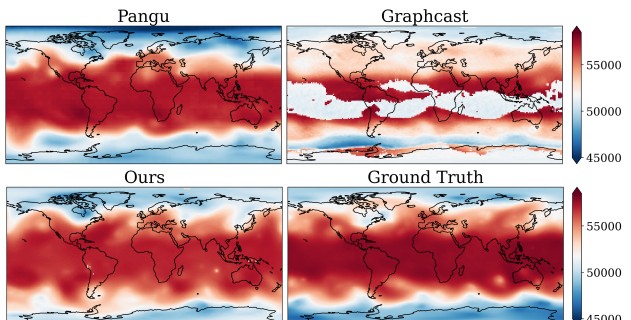

*Figure 5.* Comparison results of 100-day forecasts between the two best models and our OneForecast.

### 3.4. Extreme Events Assessment

Extreme events, such as tropical cyclones, can significantly impact human society. In this section, we evaluate our model's ability in forecasting those extreme cyclones. As

*Table 2.* Comparison results of RMSE between deterministic forecasts and ensemble forecasts (ENS), the best results are in **bold**.

| MODEL | FORECAST DAY | | | |
|---|---|---|---|---|
| | 7-DAY | 8-DAY | 9-DAY | 10-DAY |
| PANGU | 0.4875 | 0.5321 | 0.5742 | 0.6213 |
| PANGU (ENS) | 0.4435 | 0.4743 | 0.4979 | 0.5205 |
| GRAPHCAST | 0.4440 | 0.4923 | 0.5346 | 0.5823 |
| GRAPHCAST (ENS) | 0.4412 | 0.4759 | 0.5072 | 0.5331 |
| FUXI | 0.5928 | 0.6314 | 0.6604 | 0.6968 |
| FUXI (ENS) | 0.4898 | 0.5175 | 0.5353 | 0.5498 |
| ONEFORECAST | **0.4268** | 0.4834 | 0.5313 | 0.5809 |
| ONEFORECAST (ENS) | 0.4393 | **0.4699** | **0.4951** | **0.5167** |

shown in Figure 1, OneForecast achieves competitive performance in typhoon tracking during two extreme events, Yagi (2018) and Molva (2020). For Yagi, due to Best Track (Ying et al., 2014) (Lu et al., 2021) doesn't report intact track, we treat ERA5 as the ground truth. For Molva, we treat Best Track as the ground truth. The details of tracking algorithm can be found in Appendix F.3. Additionally, we download the forecast results of baseline models (e.g., Pangu, Fuxi and Graphcast) from WeatherBench2, which is trained using high resolution (0.25°) data, to better illustrate the performance of the baselines. Although OneForecast uses lower-resolution (1.5°) data, which may limit its capacity to predict cyclones, it nevertheless shows strong forecast skills in tracking tropical cyclones comparing with the baselines.

### 3.5. Long-term and Ensemble Forecasts

As shown in Figure 5, we evaluate long-term forecasts with the two best models on Z500 (500 hPa Geopotential). Pangu exhibits patch artifacts in 100-day forecasts, while GraphCast experiences error accumulation that degrades the forecasted physical fields, rendering them physically implausible over time. In contrast, OneForecast achieves stable long-term forecast performance, effectively capturing large-scale atmospheric states without the aforementioned issues.

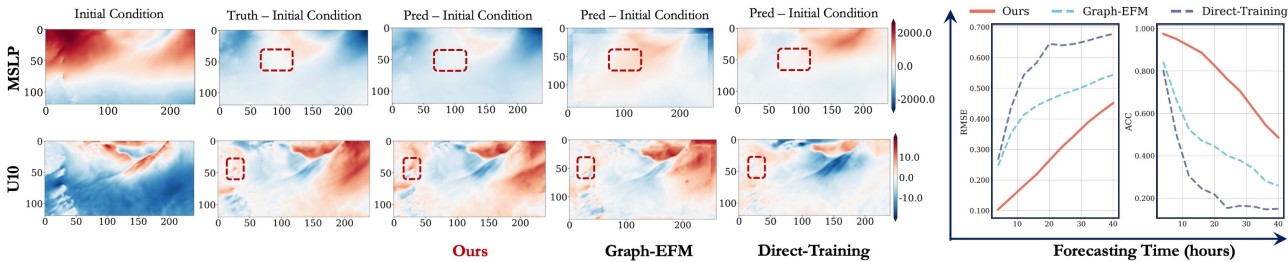

*Figure 6.* High-resolution regional results. In the left figure, we select two variables, MSLP and U10M, for visualization. We compare our model with Graph-EFM and the method that directly trains on high-resolution data. The right figure shows line charts of RMSE and ACC for different models over time. These two figures demonstrate that our proposed neural nesting grid method achieves the best performance.

These results highlight OneForecast's superior capability in maintaining accurate and physically consistent predictions over extended forecasting horizons. In Table 2, we show the results for 10 initial conditions (starting from 00:00 UTC 1 January 2020, and the interval between consecutive initial conditions is 12 hour). The ensemble forecast (ENS) results are averaged from 50 members. Obviously, in most cases (especially for longer time), the forecasting performance is enhanced for each model when uncertainty is incorporated, and OneForecast still achieves the best performance.

### 3.6. Ablation Studies

To verify the effectiveness of the proposed method, as shown in Table 3, we conduct detailed ablation experiments. We introduce the following model variants: (1) **OneForecast w/o Region Refined Graph**, we remove the region refined mesh from the finest mesh and compute the regional metrics. (2) **OneForecast w Region Refined Graph**, we reserve the region refined mesh. (3) **OneForecast R w/o NNG**, we remove the neural nested grid method (NNG) in the regional forecasts and only use the regional data to train the model. (4) **OneForecast R BF**, we remove the NNG in the regional forecasts and only use the boundary forcing method to train the model. (5) **OneForecast R w NNG**, regional forecast model with NNG. (6) **OneForecast w/o MSM**, we remove the multi-stream messaging module (MSM) and use a traditional MLP-based messaging module. (7) **OneForecast**, the full version of OneForecast for the global forecasts. For (1) and (2), we only evaluate the region-refined data. For (3), (4), and (5), we only evaluate the specific regional data. For (6) and (7), we evaluate the global data. And these results are based on 4-day forecasts for 100 ICs. Experimental results show that whether it is a global or regional forecast task, the lack of any component will degrade the performance of OneForecast, which proves the effectiveness of the proposed method. And as shown in Figure 7, the proposed MSM can better capture of high and low frequency information, which achieves satisfactory results in long-term forecasts and extreme event forecasts, such as typhoons.

*Table 3.* Ablation studies on 1.5° WeatherBench2 benchmark, the best results are in **bold**.

| Variants | RMSE | ACC |
|---|---|---|
| ONEFORECAST W/O REGION REFINED GRAPH | 0.3793 | 0.6075 |
| ONEFORECAST W REGION REFINED GRAPH | **0.2609** | **0.8099** |
| ONEFORECAST R W/O NNG | 0.5828 | 0.2450 |
| ONEFORECAST R BF | 0.4428 | 0.4711 |
| ONEFORECAST R W NNG | **0.2180** | **0.8856** |
| ONEFORECAST W/O MSM | 0.3921 | 0.9305 |
| ONEFORECAST | **0.2954** | **0.9577** |

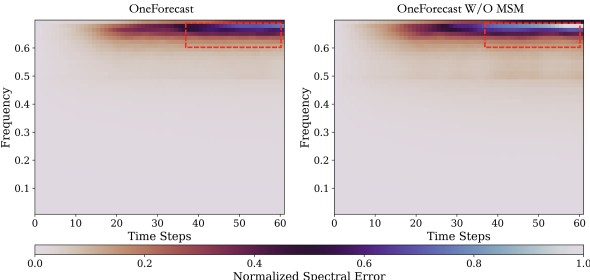

*Figure 7.* Normalized spectral error of the proposed MSM and the traditional MLP-based massaging.

## 4. Conclusion

In this paper, we propose OneForecast, a global-regional nested weather forecasting framework leveraging multi-scale graph neural networks. By integrating dynamical systems principles with multi-grid structures, our approach refines target regions for improved capture of high-frequency features and extreme events. The adaptive information propagation mechanism, featuring dynamic gating units, mitigates over-smoothing and enhances node-edge feature representation. Additionally, the proposed neural nested grid method preserves global information for regional forecasts, effectively relieves the loss of boundary information, which improves the regional forecast performance. Empirical results show that the proposed OneForecast achieves higher prediction accuracy at global and regional scales, especially for long-term and extreme event predictions, marking a step towards more robust data-driven weather forecasting.

## Acknowledgements

This work was supported by the National Natural Science Foundation of China (42125503, 42430602).

## Impact Statement

This paper presents work whose goal is to advance the field of Machine Learning. There are many potential societal consequences of our work, none which we feel must be specifically highlighted here. In addition, we propose OneForecast, a global-regional weather forecasting model based on graph neural networks. In the future, we will further study the combination of graph neural networks and dynamic systems to improve the accuracy of forecasts, especially for extreme event forecasts.

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

# A. Proofs of Theorems

**Theorem A.1** (High-pass Filtering Property of Multi-stream Messaging). *Consider an improved multi-stream message passing mechanism. Let a graph signal $\mathbf{f} \in \mathbb{R}^N$ have a spectrum $\hat{\mathbf{f}} = \mathbf{U}^\top \mathbf{f}$ under the graph Fourier basis $\mathbf{U} = [\mathbf{u}_1, \mathbf{u}_2, \ldots, \mathbf{u}_N]$, where $\mathbf{L} = \mathbf{U}\mathbf{\Lambda}\mathbf{U}^\top$ is the normalized graph Laplacian and $\mathbf{\Lambda} = \mathrm{diag}(\lambda_1, \lambda_2, \ldots, \lambda_N)$ with $0 \leq \lambda_1 \leq \cdots \leq \lambda_N \leq 2$. Define the frequency response function of the message passing operator by $\rho : \lambda \mapsto \mathbb{R}$. If the dynamic gating weights satisfy*

$$g_i^{(h,e)}, \; g_i^{(h,s)}, \; g_i^{(h,d)} \; \propto \; |\lambda_i - 1| \; + \; \epsilon \quad (\epsilon > 0), \tag{20}$$

*then there exist constants $\alpha > 0$ and $\kappa > 0$ such that the frequency response $\rho(\lambda_i)$ of the operator satisfies*

$$\rho(\lambda_i) \; \geq \; \alpha \, |\lambda_i - 1| \quad and \quad \rho(\lambda_i) \; \geq \; \kappa \, \lambda_i, \tag{21}$$

*which means the operator is a strictly high-pass filter.*

*Proof.* We consider a multi-stream message passing operator that we denote by $\mathbf{G}$. This operator depends on both node features and dynamic gating on edges. We let $\mathbf{f} \in \mathbb{R}^N$ be an arbitrary graph signal, and we write its graph Fourier transform as $\hat{\mathbf{f}} = \mathbf{U}^\top \mathbf{f}$, where $\mathbf{L} = \mathbf{U}\mathbf{\Lambda}\mathbf{U}^\top$ is the normalized Laplacian and $\mathbf{\Lambda} = \mathrm{diag}(\lambda_1, \ldots, \lambda_N)$ with $0 \leq \lambda_1 \leq \cdots \leq \lambda_N \leq 2$.
**Step 1: Integral analogy of gating on a discrete graph.** We first recall that on a continuous domain, a gating operator often admits a representation of the form

$$(\mathbf{G}\mathbf{f})(x) \; = \; \int \mathcal{K}(x, \xi) \, \mathbf{f}(\xi) \, d\xi,$$

where $\mathcal{K}(x, \xi)$ is a kernel that encodes the gating weights. On a discrete graph, the integral turns into a finite sum. Hence for node $i$, we write

$$(\mathbf{G}\mathbf{f})_i \; = \; \sum_{j \in \mathcal{N}_i} \mathcal{K}_{ij} \, f_j,$$

where $\mathcal{N}_i$ denotes the neighbors of node $i$, and $\mathcal{K}_{ij}$ depends on the dynamic gating parameters $\left(g_i^{(h,e)}, g_i^{(h,s)}, g_i^{(h,d)}\right)$. We assume these gating parameters scale proportionally to $|\lambda - 1| + \epsilon$, which implies larger weights when $\lambda$ is around high or mid-frequency regions.

**Step 2: Spectral decomposition of the operator.** We decompose $\mathbf{f}$ in the eigenbasis of $\mathbf{L}$:

$$\mathbf{f} \; = \; \sum_{\ell=1}^{N} \hat{f}_\ell \, \mathbf{u}_\ell, \quad f_j \; = \; \sum_{\ell=1}^{N} \hat{f}_\ell \, (\mathbf{u}_\ell)_j.$$

The operator $\mathbf{G}$ acts on $\mathbf{u}_\ell$ with some gain factor $\rho(\lambda_\ell)$, which we call the frequency response. In other words, we write

$$\mathbf{G}\, \mathbf{u}_\ell \; = \; \rho(\lambda_\ell) \, \mathbf{u}_\ell.$$

Thus the value of $\rho(\lambda_\ell)$ reveals how the operator scales the amplitude of the $\ell$-th eigenmode.

**Step 3: High-pass filtering behavior from gating design.** We now analyze the effect of gating weights $g_i^{(h,e)}$, $g_i^{(h,s)}$, $g_i^{(h,d)}$ that satisfy

$$g_i^{(h,e)}, \, g_i^{(h,s)}, \, g_i^{(h,d)} \; \propto \; |\lambda_i - 1| + \epsilon.$$

Because $\lambda_i \in [0, 2]$, when $\lambda_i$ is close to 2, it represents a high-frequency component on the graph. In that regime, the gating weights become larger, and the message passing operator $\mathbf{G}$ amplifies those components. Similarly, when $\lambda_i$ is near 1, the factor $|\lambda_i - 1|$ can still be significant enough to enhance mid-to-high frequencies. Conversely, for $\lambda_i$ near 0 (low frequency), the gating is relatively small and thus tends to suppress those components.

**Step 4: Combining inequalities to show strictly high-pass.** We combine partial inequalities for different ranges of $\lambda_i$. Since $0 \leq \lambda_i \leq 2$, we use the gating assumption to show there are positive constants $\alpha$ and $\kappa$ such that

$$\rho(\lambda_i) \; \geq \; \alpha \, |\lambda_i - 1| \quad and \quad \rho(\lambda_i) \; \geq \; \kappa \, \lambda_i.$$

Hence the operator $\mathbf{G}$ behaves like a high-pass filter, because it provides higher gain to higher (or mid-high) frequency components and less gain to low-frequency components. Therefore, $\mathbf{G}$ is a strictly high-pass operator in the graph spectral domain.

This completes the proof of Theorem A.1. □

# B. Related Work

## B.1. Deep Learning based Weather Forecasting

**Global Weather Forecasting.** Global weather forecasting has seen significant progress with deep learning models. FourCastNet, based on Fourier neural operators, provides global forecasts comparable to traditional numerical methods like IFS, but at much higher speeds (Pathak et al., 2022). Pangu, utilizing the Swin Transformer, exceeds NWP methods, incorporating earth-specific location embeddings for better performance (Bi et al., 2023). The Spherical Fourier Neural Operator (SFNO) extends Fourier methods using spherical harmonics, offering more stable long-term predictions (Bonev et al., 2023). FuXi focuses on long-term forecasting, achieving a 15-day forecasts comparable to ECMWF (Chen et al., 2023b). GraphCast leverages message-passing networks to improve efficiency and forecasting accuracy (Lam et al., 2023), and GenCast builds on this to enhance ensemble forecasting (Price et al., 2023). Further, diffusion models like those in (Li et al., 2024) generate probabilistic ensembles by sampling, while NeuralGCM (Kochkov et al., 2024) focuses on atmospheric circulation with a dynamic core, offering climate simulation capabilities but at higher training and inference costs.

**Regional Weather Forecasting.** The goal of regional weather forecasting is to enhance local prediction accuracy with high-resolution models. CorrDiff (Mardani et al., 2023) combines U-Net and diffusion models to improve local forecasts. MetaWeather (Kim et al., 2024) adapts global forecasts to regional contexts using meta-learning. GNNs are also widely applied in regional forecasting, with Graphcast (Lam et al., 2023) enhancing accuracy by modeling complex spatial dependencies. MetNet-3 (Espeholt et al., 2022) offers high-accuracy forecasts for weather variables, such as precipitation, temperature, and wind speed, at 2-minute intervals and 1–4 km resolution, outperforming traditional models like HRRR. NowcastNet (Zhang et al., 2023) and DGMR (Ravuri et al., 2021) excel in short-term extreme precipitation forecasts using deep generative models and radar data. In spatiotemporal prediction, NMO (Wu et al., 2024b) models the evolution of physical dynamics, providing new insights for local weather forecasting. Similarly, SimVP (Gao et al., 2022) and PastNet (Wu et al., 2024c) achieve good results in forecasting local precipitation evolution using spatiotemporal convolution methods.

## B.2. Numerical analysis methods

Multigrid methods (McCormick, 1987; Wesseling, 1995; Hackbusch, 2013; Bramble, 2019; Hiptmair, 1998; Brandt et al., 1983; Borzi & Schulz, 2009) and nested grid strategies (Miyakoda & Rosati, 1977; Zhang et al., 2012; Sullivan et al., 1996) are widely used to solve PDEs and handle multi-scale problems (Debreu & Blayo, 2008; Xue et al., 2000). Multigrid methods use grids of different resolutions to transfer information and accelerate iterations. They efficiently solve large-scale problems and improve computational accuracy. By eliminating low-frequency errors on coarse grids and high-frequency errors on fine grids, multigrid methods effectively handle error convergence at different scales (He & Xu, 2019; He et al., 2023; Shao et al., 2022). Nested grid strategies embed higher-resolution fine grids into regions of interest based on a global coarse grid to capture local complex physical phenomena in detail. In weather forecasting, this method provides large-scale background fields on a global scale while refining the grid for target regions to accurately simulate the evolution of local weather systems and the occurrence of extreme events (Bacon et al., 2000).

# C. Data Details

## C.1. Dataset

In this section, we are going to introduce the dataset we used in this study detailedly. For the global forecasting, we conduct experiments on the WeatherBench2 (Rasp et al., 2024) benchmark, a subset of ERA5 reanalysis data (Hersbach et al., 2020). The WeatherBench2 benchmark we used is the version of 1.5° resolution ($121 \times 240$), which spans from 1959 to 2020. This subset contains 5 variables (Z, Q, T, U, V) with 13 pressure levels (50 hPa, 100 hPa, 150 hPa, 200 hPa, 250 hPa, 300 hPa, 400 hPa, 500 hPa, 600 hPa, 700 hPa, 850 hPa, 925 hPa and 1,000 hPa) and 4 variables (U10M, V10M, T2M, MSLP) with surface level. For the regional forecasting, a higher resolution data (0.25° resolution) of ERA5 is also used, which can be downloaded from `https://cds.climate.copernicus.eu/`, the official website of Climate Data Store (CDS). All the data we used are shown in Table 4. For both global and regional forecasts, we use the data from 1959 to 2017 for training, 2018 to 2019 for validating, and 2020 for testing.

Table 4. The data details in this work.

| TASK | VARIABLE NAME | LAYERS | SPATIAL RESOLUTION | DT | LAT-LON RANGE | TIME |
|------|---------------|--------|--------------------|-----|---------------|------|
| GLOBAL | GEOPOTENTIAL (Z) | 13 | 1.5° | 6H | -90°S-180°W∼90°N180°E | 1959∼2020 |
|  | SPECIFIC HUMIDITY (Q) | 13 | 1.5° | 6H | -90°S-180°W∼90°N180°E | 1959∼2020 |
|  | TEMPERATURE (T) | 13 | 1.5° | 6H | -90°S-180°W∼90°N180°E | 1959∼2020 |
|  | U COMPONENT OF WIND (U) | 13 | 1.5° | 6H | -90°S-180°W∼90°N180°E | 1959∼2020 |
|  | V COMPONENT OF WIND (V) | 13 | 1.5° | 6H | -90°S-180°W∼90°N180°E | 1959∼2020 |
|  | 10 METRE U WIND COMPONENT (U10M) | 1 | 1.5° | 6H | -90°S-180°W∼90°N180°E | 1959∼2020 |
|  | 10 METRE V WIND COMPONENT (V10M) | 1 | 1.5° | 6H | -90°S-180°W∼90°N180°E | 1959∼2020 |
|  | 2 METRE TEMPERATURE (T2M) | 1 | 1.5° | 6H | -90°S-180°W∼90°N180°E | 1959∼2020 |
|  | MEAN SEA LEVEL PRESSURE (MSLP) | 1 | 1.5° | 6H | -90°S-180°W∼90°N180°E | 1959∼2020 |
| REGIONAL | MEAN SEA LEVEL PRESSURE (MSLP) | 1 | 0.25° | 6H | 7.5°W114°E∼ 36°W172.5°E | 1959∼2020 |
|  | 10 METRE U WIND COMPONENT (U10M) | 1 | 0.25° | 6H | 7.5°W114°E∼ 36°W172.5°E | 1959∼2020 |

Table 5. The Params and MACs comparsion of different models.

| MODEL | PARAMS (M) | MACs (G) |
|-------|------------|----------|
| PANGU (BI ET AL., 2023) | 23.83 | 142.39 |
| FENGWU (CHEN ET AL., 2023A) | 153.49 | 132.83 |
| GRAPHCAST (LAM ET AL., 2023) | 28.95 | 1639.26 |
| FUXI (CHEN ET AL., 2023B) | 128.79 | 100.96 |
| ONEFORECAST | 24.76 | 509.27 |

## C.2. Data preprocessing

Different atmosphere and ocean variables have large variations in their magnitude. To allow the model focusing on predictions rather than learning the differences between variables, we normalized the data before putting the data into the model. We calculated the mean and standard deviation of all variables using data from 1959 to 2017 (training set). Each variable has a corresponding mean and standard deviation. Before feeding the data into the model, we first subtract the respective mean and divided it by the standard deviation.

## D. Algorithm

We summarize the overall framework of OneForecast in Algorithm 1.

## E. Model Details for Global Forecasts

### E.1. Earth-specific Region Refined Multi-scale Graph

The graph used in OneForecast can be defined as: $\mathcal{G}(\mathcal{V}^G, \mathcal{V}, \mathcal{E}, \mathcal{E}^{\text{G2M}}, \mathcal{E}^{\text{M2G}})$.

**Grid Nodes.** $\mathcal{V}^G$ is the ensemble of grid nodes, which contains $120 \times 240 = 28800$ nodes for 1.5° global data in global forecast task. And each node consists of 69 atmospheric features (5 variables at 13 pressure levels and 4 variables at surface level, $5 \times 13 + 4 = 69$). Since we just consider 1 step historical state, the input features of OneForecast are 69. For regional forecast, the region size can be arbitrary within the permission of GPU memory. For simplicity, we choose the region size of $120 \times 240$ from 0.25° data, the node is still $120 \times 240 = 28800$.

**Mesh Nodes.** $\mathcal{V}$ is the ensemble of mesh nodes, which contains multi-scale mesh nodes of different fineness and region refined mesh nodes that cover the global area. The mesh nodes are distributed over a refined icosahedron that has undergone five levels of subdivision, and the coarsest icosahedron consists of 12 vertices and 20 triangular faces. By dividing each triangular face into four smaller triangles, an additional node is generated at the midpoint of each edge. The new nodes are then projected back onto the unit sphere, gradually refining the grid. To enhance the forecasting performance in key regions, we further refine specific areas of the finest mesh, achieving localized mesh densification. For the global forecast task, we refine the 2 areas: 0°N105°E∼30°N160°E and 10°N-95°W∼30°N-35°W. The features of each node include the

---

**Algorithm 1** OneForecast Framework for Global Weather Forecasting

---

**Require:** Initial atmospheric condition $Z_t$.
**Ensure:** Next step atmospheric state $Z_{t+1}$.
 1: Initialize OneForecast
 2: **repeat**
 3:     **Encoder**
 4:     Embedding features of grid nodes $Z_t$, mesh nodes $\mathcal{V}$, mesh edges $\mathcal{E}$, grid to mesh edges $\mathcal{E}^{\mathrm{G2M}}$, and mesh to grid edges $\mathcal{E}^{\mathrm{M2G}}$ into latent space using respective MLP: $(\mathcal{V}_f^G, h, \mathcal{E}_f, \mathcal{E}_f^{\mathrm{G2M}}, \mathcal{E}_f^{\mathrm{M2G}}) = \mathrm{MLPs}(Z_t, \mathcal{V}, \mathcal{E}, \mathcal{E}^{\mathrm{G2M}}, \mathcal{E}^{\mathrm{M2G}})$
 5:     Project the atmospheric state from the lat-lon grid into the mesh nodes: $\mathcal{E}_f^{\mathrm{G2M}'} = \mathrm{ESMLP}(\mathcal{V}_f^G, h, \mathcal{E}_f^{\mathrm{G2M}})$,
     $h' = \mathrm{MLP}_{e1}(h, \sum \mathcal{E}_f^{\mathrm{G2M}'})$
 6:     Update grid node feature: $\mathcal{V}_f^{G'} = \mathrm{MLP}_{e2}(\mathcal{V}_f^G)$
 7:     Apply residual connection to update the feature of grid to mesh edge, mesh node, and grid node again: $\mathcal{E}_f^{\mathrm{G2M}} = \mathcal{E}_f^{\mathrm{G2M}'} + \mathcal{E}_f^{\mathrm{G2M}}, h' = h' + h, \mathcal{V}_f^G = \mathcal{V}_f^{G'} + \mathcal{V}_f^G$
 8:     **Multi-stream Messaging**
 9:     Apply dynamic multi-head gated edge update module (DMG) to update edge feature: $\mathcal{E}_f' = DMG(\mathcal{E}_f, h_s, h_r)$
10:     Apply multi-head node attention mechanism (MHA) to update mesh node feature: $h' = MHA(h, \sum \mathcal{E}_f')$
11:     Apply residual connection to update the feature of edge and mesh node: $\mathcal{E}_f = \mathcal{E}_f' + \mathcal{E}_f, h = h' + h$
12:     **Decoder** Project the feature from mesh back to lat-lon grid: $\mathcal{E}_f^{\mathrm{M2G}'} = \mathrm{ESMLP}(\mathcal{V}_f^G, \mathcal{E}_f, \mathcal{E}^{\mathrm{M2G}})$, $\mathcal{V}_f^{G'} = \mathrm{MLP}_{d1}(\mathcal{V}_f^G, \sum \mathcal{E}_f^{\mathrm{M2G}'}), \mathcal{V}_f^G = \mathcal{V}_f^G + \mathcal{V}_f^{G'}, Z_{t+1} = \mathrm{MLP}_{d2}(\mathcal{V}_f^G)$
13: **until** converged
14: **return** $OneForecast$

---

cosine value of the latitude, as well as the sine and cosine values of the longitude. We only keep the finest mesh nodes, since the nodes on the coarse mesh are its subset. In total, the graph structure of OneForecast comprises 12337 mesh nodes, each characterized by three features.

**Mesh Edges.** $\mathcal{E}$ are the bidirectional edges that connect mesh nodes (sender and receiver nodes). Similar to mesh nodes, there are corresponding edges for each scale of mesh, and $\mathcal{E}$ is the ensemble of multi-scale edges. And the features of each edge include the length of edge, the 3D position difference between sender and receiver nodes. In total, OneForecast comprises 98296 mesh edges, each characterized by four features.

**Grid2Mesh Edges.** $\mathcal{E}^{\mathrm{G2M}}$ are the the unidirectional edges that used in the encoder, which connect grid and mesh nodes. To ensure that each grid node has a corresponding mesh node connected to it, we add $\mathcal{E}^{\mathrm{G2M}}$ to grid nodes and mesh nodes if the distance between them is less than or equal to 0.6 times the edge length of the finest $\mathcal{E}$. Similar to mesh edge $\mathcal{E}$, each grid2mesh edge comprises 4 features, and OneForecast has 49233 grid2mesh edges in total.

**Mesh2Grid Edges.** $\mathcal{E}^{\mathrm{M2G}}$ are the unidirectional edges that used in the decoder, which connect grid and mesh nodes. For each grid node, we find the triangle face containing it on the finest mesh and connect 3 mesh nodes to it. Similar to other edges, each mesh2grid edge has 4 features. In total, OneForecast has 86,400 mesh2grid edges.

### E.2. Encoder

This paper uses 2 types MLP. We denote the first type as MLP$(\cdot)$, the number of layer is 1, the latent dim is 512, and followed by the SiLU activation function and Layernorm function. And we denote the second type as ESMLP$(\cdot)$, the other hyperparameters are the same as MLP$(\cdot)$, except ESMLP$(\cdot)$ transforms three features (edge features, node features of the corresponding source and destination node) individually through separate linear transformations and then sums them for each edge accordingly. We first apply embedder MLP to map the data to the latent space, which can be defined as:

$$MLP = LN(SiLU(Linear(x_{embedder}))), \tag{22}$$

$$(\mathcal{V}_f^G, h, \mathcal{E}_f, \mathcal{E}_f^{\mathrm{G2M}}, \mathcal{E}_f^{\mathrm{M2G}}) = MLPs(Z_t, \mathcal{V}, \mathcal{E}, \mathcal{E}^{\mathrm{G2M}}, \mathcal{E}^{\mathrm{M2G}}), \tag{23}$$

where, $x_{embedder}$ is the input of embedder MLP. For the linear function, we set the latent dim to 512. $SiLU(\cdot)$ is the SiLU activation function, $LN(\cdot)$ is the layernorm function. $Z_t, \mathcal{V}, \mathcal{E}, \mathcal{E}^{\mathrm{G2M}}$, and $\mathcal{E}^{\mathrm{M2G}}$ are embedded features of grid nodes, mesh

nodes, mesh edges, grid to mesh edges, and mesh to grid edges. We then project the atmospheric state from the lat-lon grid into the mesh nodes. Specifically, we first update the edge features through an Edge Sum MLP (ESMLP):

$$\mathcal{E}_f^{\text{G2M}'} = \mathbf{W}_e \mathcal{E}_f^{\text{G2M}}, \tag{24}$$

$$h_s{}' = \mathbf{W}_s h_s, \tag{25}$$

$$h_d' = \mathbf{W}_d h_d + \mathbf{b}_d, \tag{26}$$

$$\mathbf{h}_{\text{sum}} = \mathcal{E}_f^{\text{G2M}'} + h_s' + h_d', \tag{27}$$

$$\mathcal{E}_f^{\text{G2M}'} = \text{LN}\left(\mathbf{W}_{\text{ESMLP}} \sigma\left(h_{sum}\right) + \mathbf{b}_{\text{ESMLP}}\right), \tag{28}$$

where, $\mathbf{W}_e$, $\mathbf{W}_s$, $\mathbf{W}_d$ are the linear transformation matrix of grid2mesh edge features, send node feature, and target node features. $\mathbf{W}_{\text{ESMLP}}$ is the linear transformation matrix of output layer. $b_d$ is the bias of mesh node during linear transformation, $b_{ESMLP}$ is the bias vector of ESMLP. In summary, the grid2mesh edge update process can be define as:

$$\mathcal{E}_f^{\text{G2M}'} = \text{ESMLP}(\mathcal{E}_f^{\text{G2M}}, h_s, h_d). \tag{29}$$

After updating the grid2node features, we update the mesh node features using another MLP:

$$h' = \text{MLP}_{\text{e1}}(h, \sum \mathcal{E}_f^{\text{G2M}'}), \tag{30}$$

where, $\sum \mathcal{E}_f^{\text{G2M}'}$ are the edges that arrives at mesh node. Then, we update the grid node features using another MLP:

$$\mathcal{V}_f^{G'} = \text{MLP}_{e2}\left(\mathcal{V}_f^{G}\right). \tag{31}$$

Finally, residual connections are applied to update the feature of grid to mesh edge, mesh node, and grid node again.

### E.3. Multi-stream Messaging

The proposed multi-stream messaging is implemented by an adaptive messaging mechanism, which contains a dynamic multi-head gated edge update module and a multi-head node attention module. This part has been introduced in detail in the main text, so we only added the hyperparameter settings here. For the dynamic multi-head gated edge update module, the dimensions of the gating vector are set to 64. In the multi-head node attention module, the $MLP_a$ used to calculate the attention score consists of a linear layer, a SiLU activation function, a linear layer, and a Sigmoid function. The hidden dimension of the linear layer is 64.

### E.4. Decoder

In the decoder, we map the feature from mesh back to lat-lon grids, similar to encoder, we first update the mesh2grid features:

$$\mathcal{E}_f^{\text{M2G}'} = \text{ESMLP}\left(\mathcal{V}_f^{G}, h, \mathcal{E}_f^{\text{M2G}}\right). \tag{32}$$

Then, we update the grid node features:

$$\mathcal{V}_f^{G'} = \text{MLP}_{d1}(\mathcal{V}_f^{G}, \sum \mathcal{E}_f^{\text{M2G}'}), \tag{33}$$

where, $\sum \mathcal{E}_f^{\text{M2G}'}$ are the edges that arrives at grid node.

After that, a residual connection is applied to update the grid node features again:

$$\mathcal{V}_f^{G} = \mathcal{V}_f^{G} + \mathcal{V}_f^{G'}. \tag{34}$$

Finally, we apply a MLP to predict the next step results:

$$Z_{t+1} = \text{MLP}_{d2}\left(\mathcal{V}_f^{G}\right). \tag{35}$$

# F. Experiments Details

## F.1. Evaluation Metric

We utilize four metrics, RMSE (Root Mean Square Error) and ACC (Anomalous Correlation Coefficient), CSI (Critical Success Index), and SEDI (Symmetric Extremal Dependence Index) to evaluate the forecasting performance, which can be defined as:

$$RMSE(k,t) = \sqrt{\frac{\sum_{i=1}^{N_{\text{lat}}} \sum_{j=1}^{N_{\text{lon}}} L(i) \left( \hat{\mathbf{A}}_{ij,t}^{k} - \mathbf{A}_{ij,t}^{k} \right)^2}{N_{\text{lat}} \times N_{\text{lon}}}}, \tag{36}$$

$$\text{ACC}(k,t) = \frac{\sum_{i=1}^{N_{\text{lat}}} \sum_{j=1}^{N_{\text{lon}}} L(i) \hat{\mathbf{A}}'^{k}_{ij,t} \mathbf{A}'^{k}_{ij,t}}{\sqrt{\sum_{i=1}^{N_{\text{lat}}} \sum_{j=1}^{N_{\text{lon}}} L(i) \left( \hat{\mathbf{A}}'^{k}_{ij,t} \right)^2 \times \sum_{i=1}^{N_{\text{lat}}} \sum_{j=1}^{N_{\text{lon}}} L(i) \left( \mathbf{A}'^{k}_{ij,t} \right)^2}}, \tag{37}$$

where $\mathbf{A}_{i,j,t}^{v}$ represents the value of variable v at horizontal coordinate $(i,j)$ and time t. Latitude-dependent weights are defined as $L(i) = N_{\text{lat}} \times \frac{\cos \phi_i}{\sum_{i'=1}^{N_{\text{lat}}} \cos \phi_{i'}}$, where $\phi_i$ is the latitude at index i. The anomaly of $A$, denoted as $A'$, is computed as the deviation from its climatology, which corresponds to the long-term mean of the meteorological state estimated from 59 years of training data. To evaluate model performance, RMSE and ACC are averaged across all time steps and spatial coordinates, providing summary statistics for variable $k$ at a given lead time $\Delta t$.

$$\text{CSI}(k,t) = \frac{\text{TP}}{\text{TP} + \text{FP} + \text{FN}}, \tag{38}$$

$$\text{SEDI}(k,t) = \frac{\log(F) - \log(H) - \log(1-F) + \log(1-H)}{\log(F) + \log(H) + \log(1-F) + \log(1-H)}, \tag{39}$$

where, true positives (TP) indicate the number of cases in which the state is accurately simulated. False positives (FP) and false negatives (FN) are defined in a similar manner. The false alarm rate is denoted as $F = \frac{\text{FP}}{\text{FP}+\text{TP}}$, while the hit rate is represented as $H = \frac{\text{TP}}{\text{TP}+\text{FN}}$.

## F.2. Model Training

For the first type comparison, we train baseline models and OneForecast using the same training framework. We set the total model training epochs to 200, the initial learning rate is 1e-3, and use the cosine annealing scheduler to adjust the learning rate until the model converged. The model codes of Pangu and Graphcast we used are released by NVIDIA modulus (https://github.com/NVIDIA/modulus). The model code of Fuxi is obtained by sending an email to the author. For all models, we select the checkpoint that performed best on the validation set for comparative analysis.

## F.3. Typhoon Tracking

To track the eye of a tropical cyclone, we follow (Bi et al., 2023; Magnusson et al., 2021) to find the local minimum of mean sea level pressure (MSLP). The time step of forecast lead time is set to be 6 hours. Specifically, once the initial position of the cyclone eye is provided, we iteratively search for a local minimum of MSLP that meets the following criteria:

- There is a maximum 850hPa relative vorticity greater than $5 \times 10^{-5}$ within a 278km radius (in the Northern Hemisphere).

- There is a maximum thickness between 850hPa and 200hPa within a 278km radius when the cyclone is extratropical.

- The maximum 10m wind speed exceeds 8m/s within a 278km radius when the cyclone is over land.

Once the cyclone eye is identified, the tracking algorithm continues to find the next position within a 445km vicinity.

This study focuses on two extreme cyclones: Tropical Storm Yagi and Severe Typhoon Molave. The Yagi formed near Iwo Jima, Japan on August 6, 2018, and landed over Wenling, China on August 12. The Molave formed on October 11, 2020, and landed over the Philippines on October 25, 2020. The initial conditions for these two cyclones are set

at 0:00 UTC, August 6, 2018 and 0:00 UTC, 11 October 2020, respectively. Since there is no Fuxi results in 2018 on WeatherBench2, we can not compare it with Yagi. The results of Best Track (Ying et al., 2014) (Lu et al., 2021) can be found in https://tcdata.typhoon.org.cn/en/.

# G. Additional Results

### G.1. Efficiency Analysis

As shown in Table 5, our OneForecast has a competitive performance for Parameters and MACs. For the MACs, the size of input tensor is set to (1, 69, 120, 240). Not that for the ML-based weather forecasts, the computational cost is less important compared with the forecasting accuracy because the ML-basd model is several orders of magnitude faster (maybe tens of thousands of times) than traditional numerical methods. For instance, in numerical forecasting, a single simulation for a 10-day forecasts can take hours of computation in a supercomputer that has hundreds of nodes. In contrast, ML-based weather forecasting models just need a few seconds or minutes to produce 10-day forecasts using only 1 GPU.

### G.2. Spectral Analysis

As shown in Figure 8, we compute the surface kinetic energy spectrum and Q700 spectrum for baseline models using WeatherBench2's official results (averaged across the first 700 ICs). Our OneForecast model achieves comparable performance in this standardized evaluation framework. Notably, as Q700 data for Fuxi were not available in the WeatherBench2, only its surface kinetic energy spectrum could be analyzed.

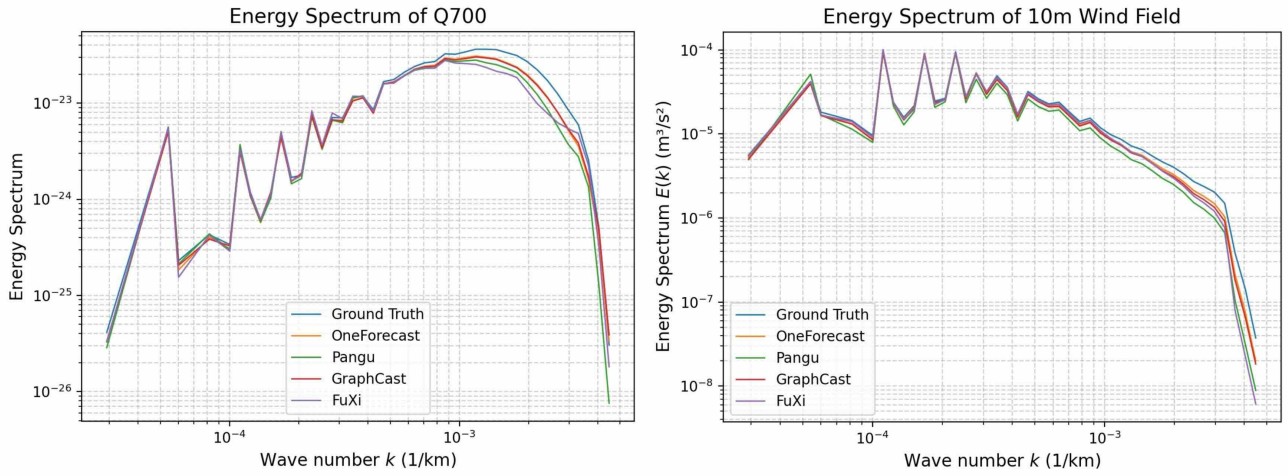

*Figure 8.* Spectral analysis of different models.

### G.3. Quantitative analysis of Extreme Event

To assess the forecast performance of more extreme events, as shown in Table 6, we present 2 extreme event assessment indicators (the higer the better) CSI and SEDI. And we also add a quantitative metric for typhoon prediction in Table 7, a lower value represents better results. It can be seen that our OneForecast also achieves satisfactory results in quantitative analysis.

### G.4. Additional quantitative comparison with the results from WeatherBench2

As shown in Figure 9, we compare all models released by WeatherBench2, except for ENS (ensemble forecasting, not the same task) and Spherical CNN (too few ICs, only 178 compared with our used 700). While the WeatherBench2 baseline leverages numerous training strategies, we only conducted 1-epoch of finetuning (Ours_finetune) during the brief rebuttal period. Nevertheless, a 2-epoch finetune model (Ours_finetune2) demonstrates improved results, indicating the potential for further gains with additional finetuning. If we finetuned for more epochs, OneForecast can achieve better result. However, our primary objective is to introduce a novel paradigm for global and regional weather forecasting rather than solely optimizing metrics, we just finetune for a few epoch as an example.

*Table 6.* Quantitative analysis of extreme event for different models.

| MODEL | WIND10M | WIND10M | T2M | T2M |
|---|---|---|---|---|
| | CSI | SEDI | CSI | SEDI |
| PANGU | 0.11 | 0.29 | 0.16 | 0.34 |
| GRAPHCAST | 0.13 | 0.29 | 0.20 | 0.38 |
| FUXI | 0.11 | 0.20 | 0.19 | 0.27 |
| OURS | 0.14 | 0.31 | 0.21 | 0.40 |

*Table 7.* Quantitative analysis of typhoon for different models.

| MODEL | TRACK POSITION ERROR(KM) |
|---|---|
| FS-HRES | 332 |
| PANGU 1.5° | 222 |
| GRAPHCAST 1.5° | 212 |
| PANGU | 231 |
| GRAPHCAST | 197 |
| OURS | 157 |

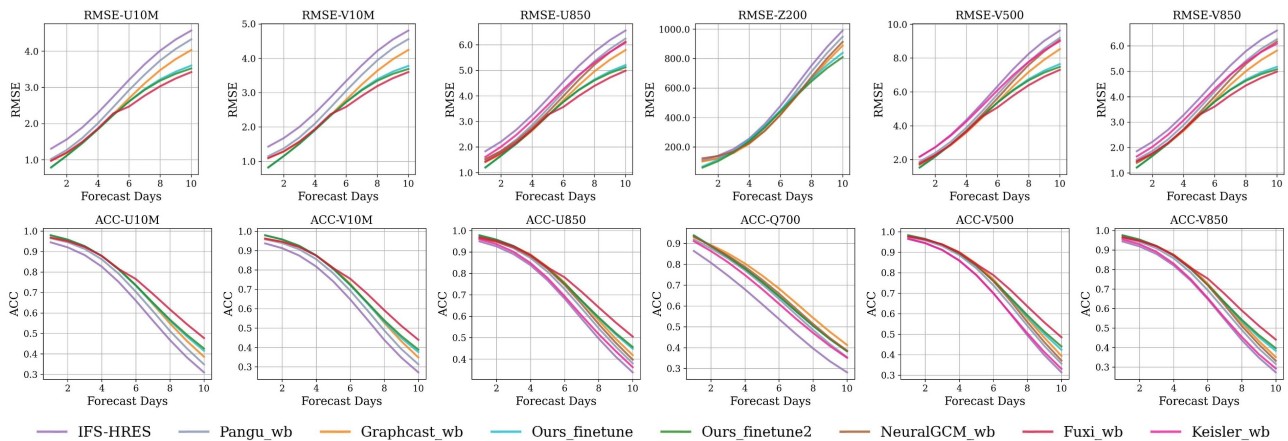

*Figure 9.* Additional quantitative comparison with the results from WeatherBench2 for several important variables.

## G.5. Additional Visual Results

We present more additional results in Figure , including 18 variables that are importrant to weather forecasting, each with results ranging from 6 hours to 10 days. These additional results further demonstrate the effectiveness of OneForecast. Same as the Figure 3, the initial conditions is 00:00 UTC, 1 January 2020.

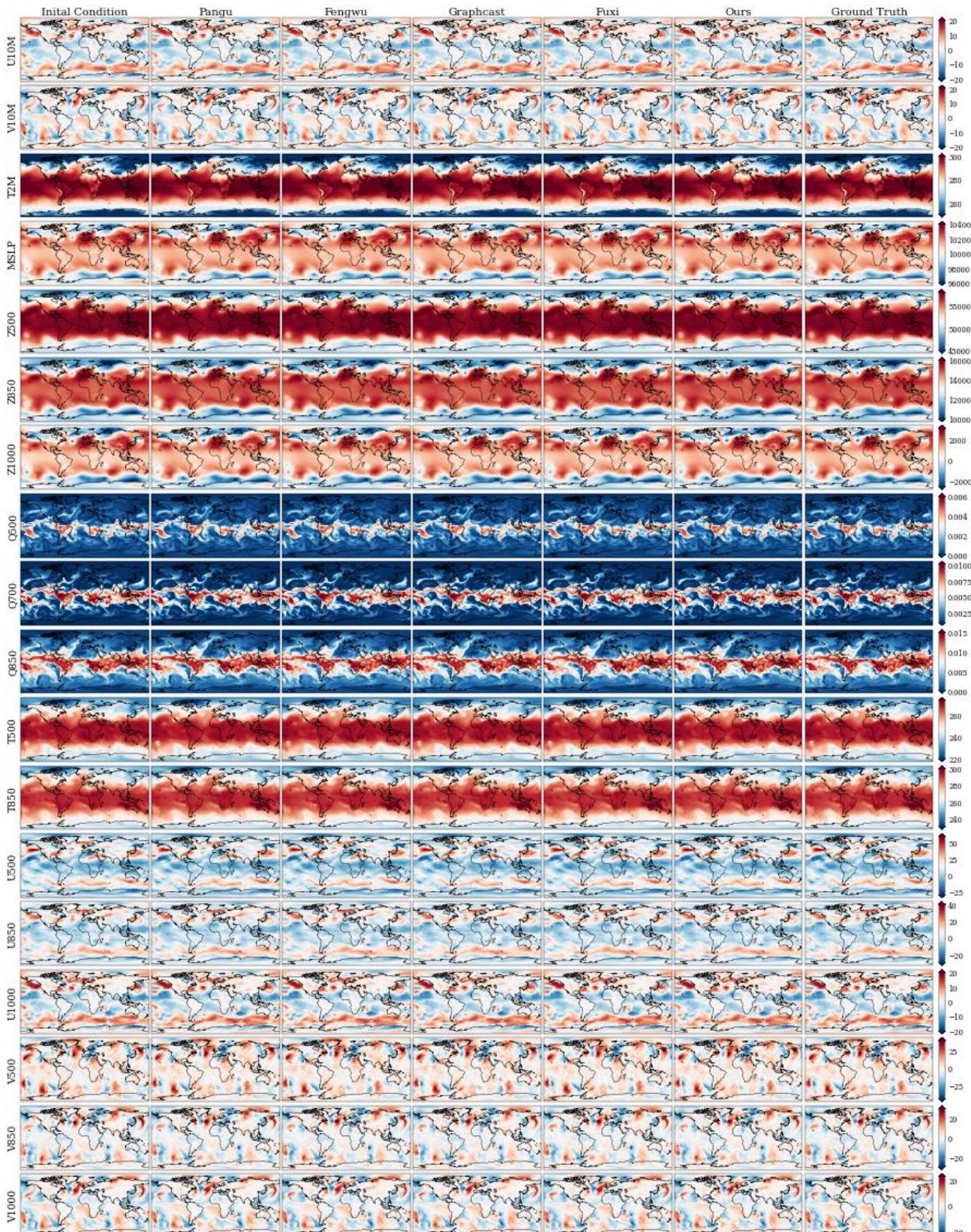

*Figure 10.* 6-hour forecast results of different models.

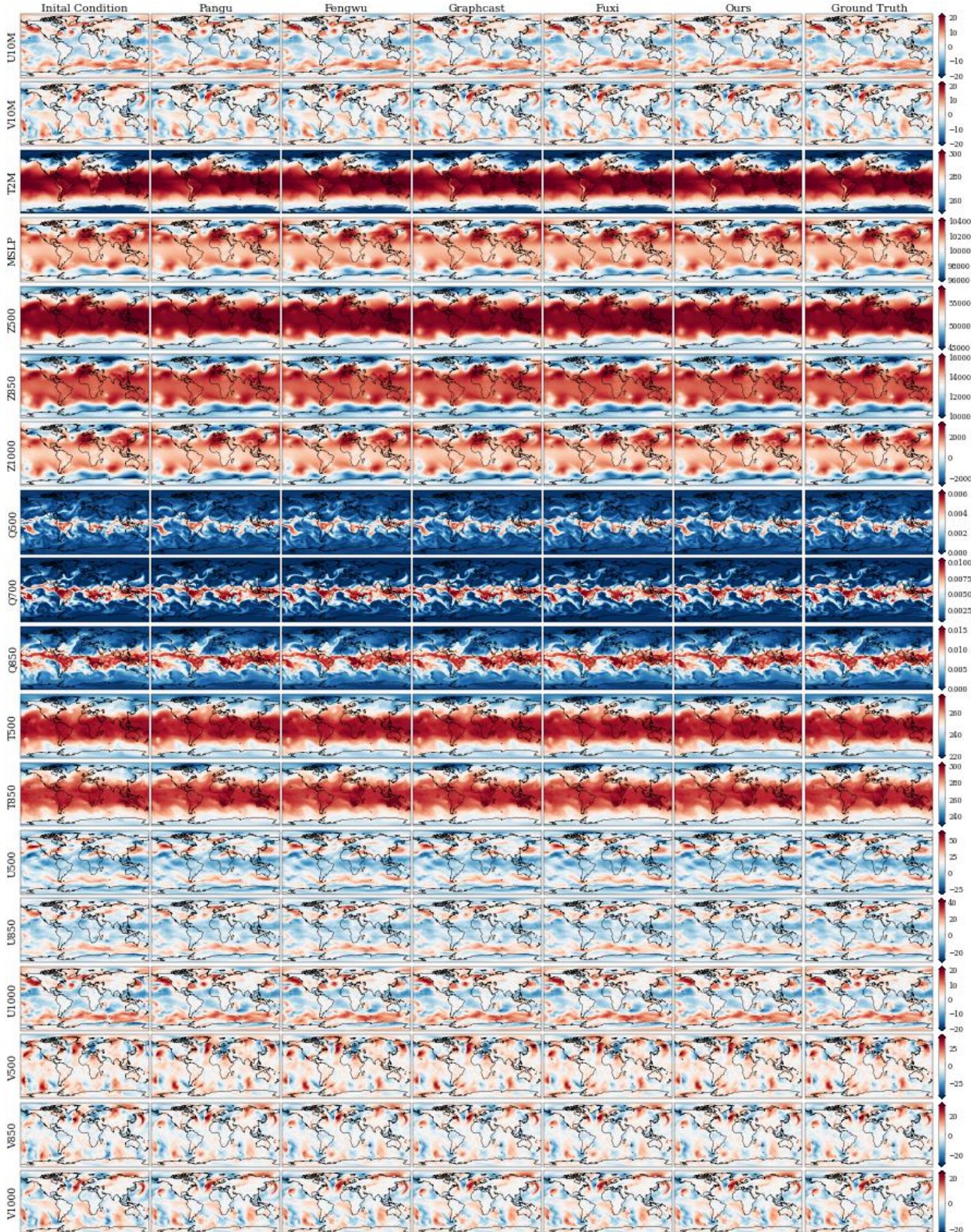

*Figure 11.* 0.5-day forecast results of different models.

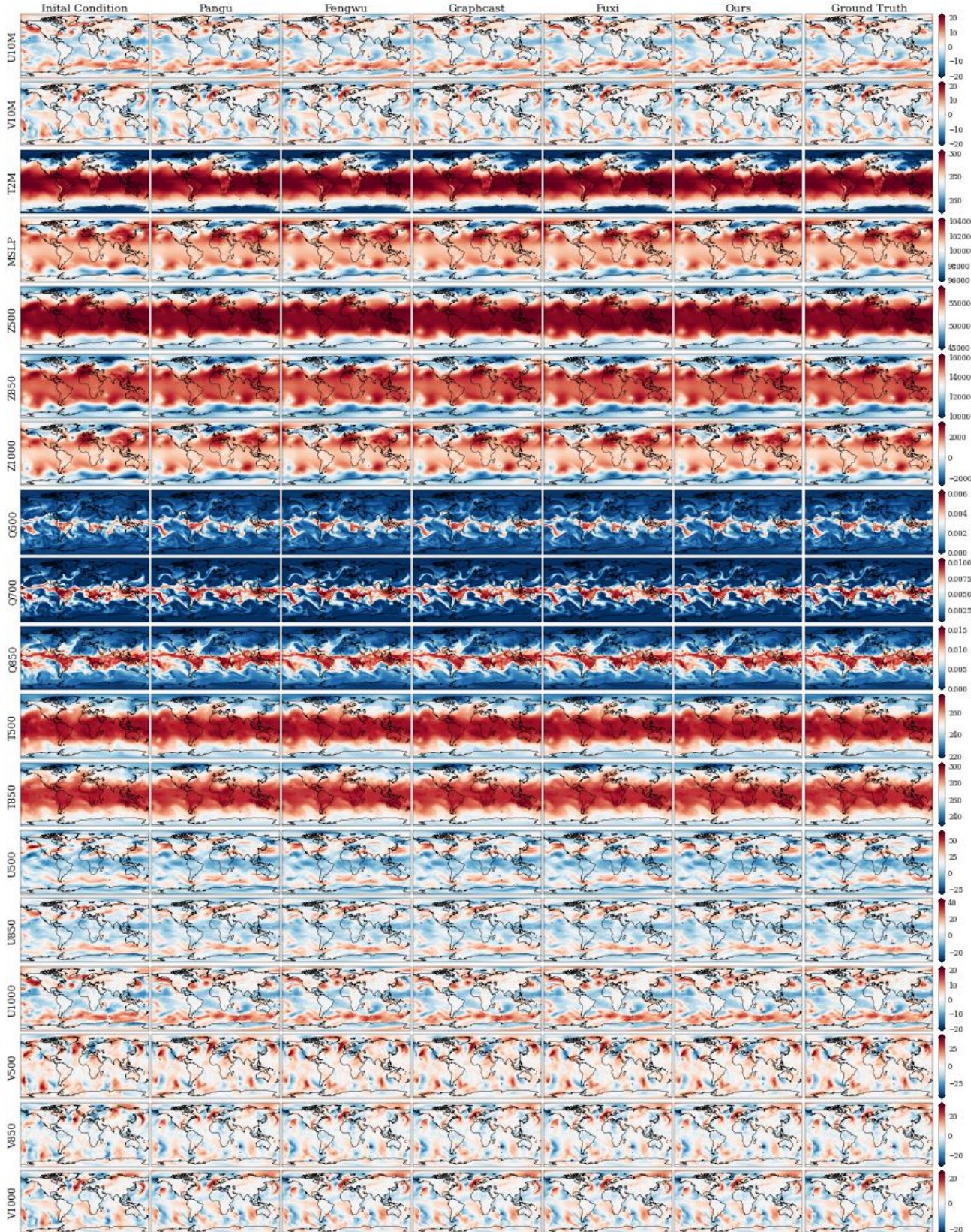

*Figure 12.* 1-day forecast results of different models.

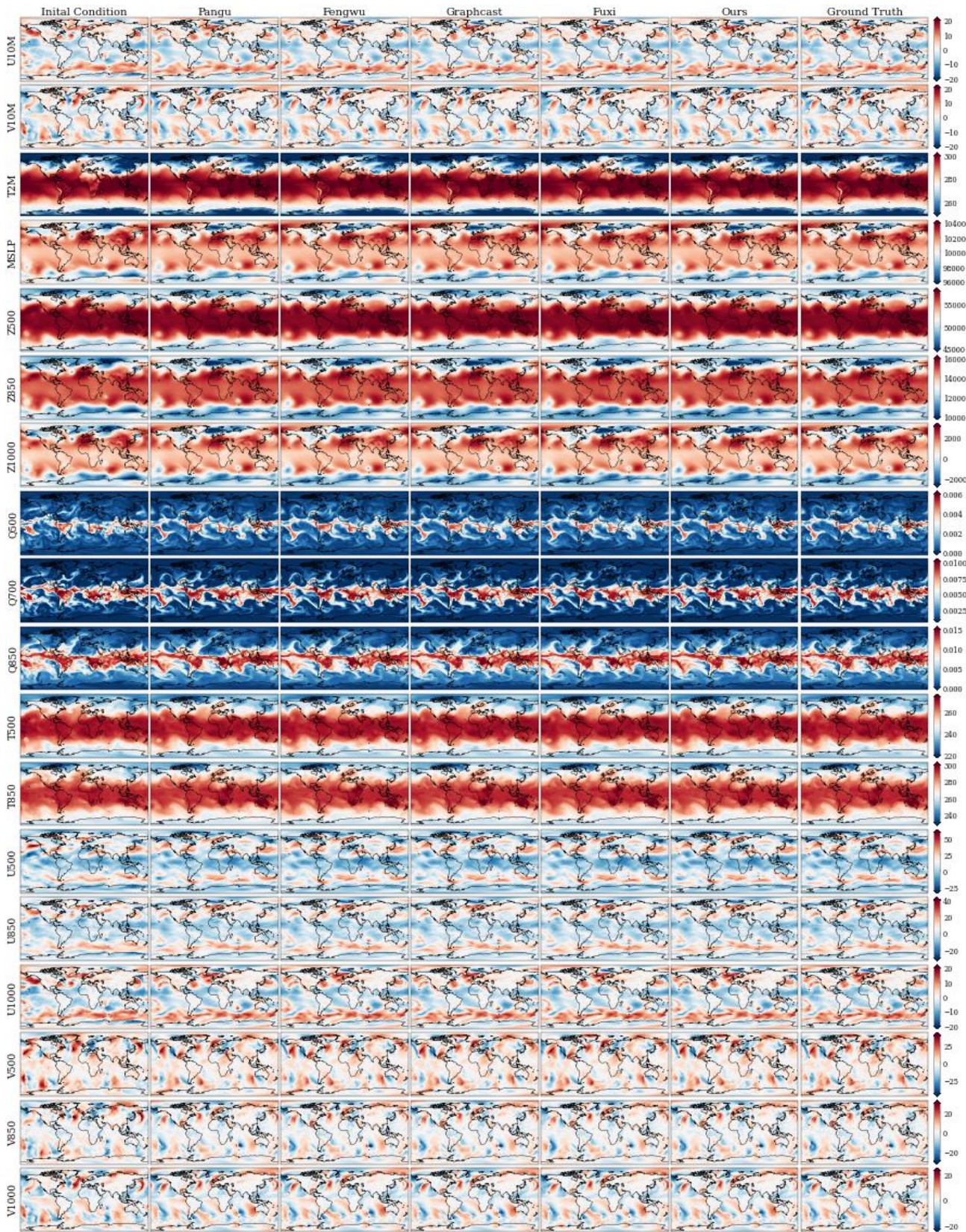

*Figure 13.* 1.5-day forecast results of different models.

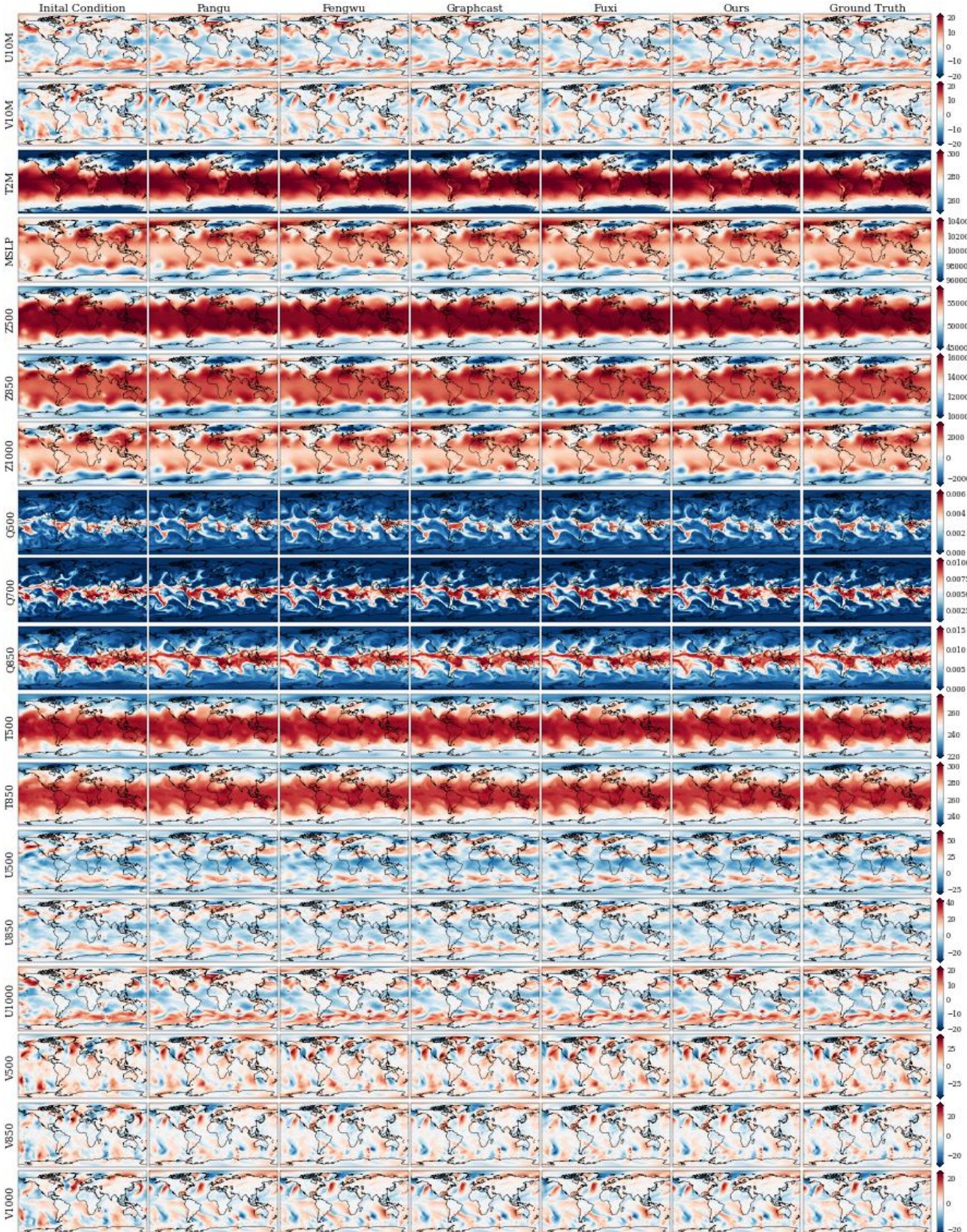

*Figure 14.* 2-day forecast results of different models.

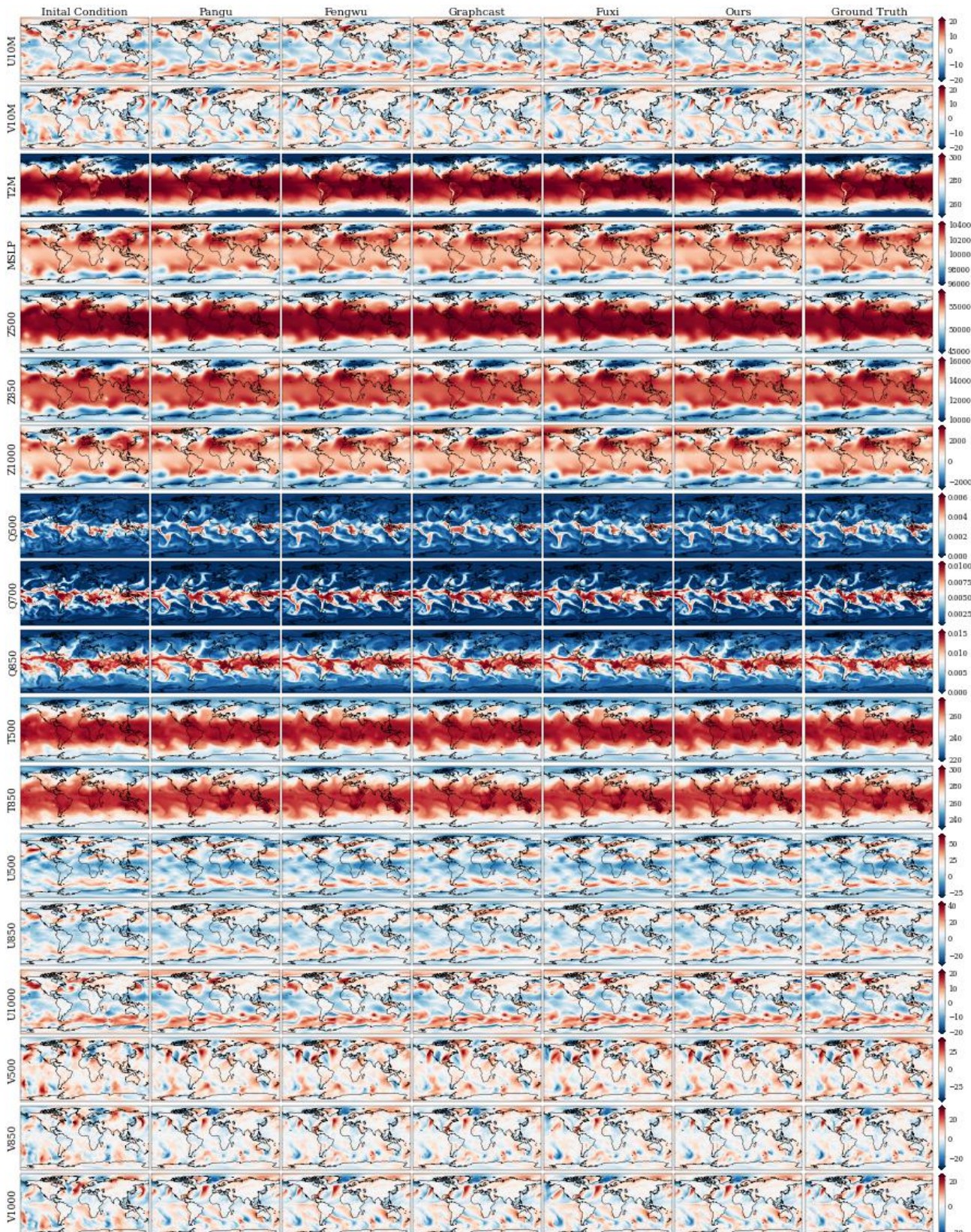

*Figure 15.* 2.5-day forecast results of different models.

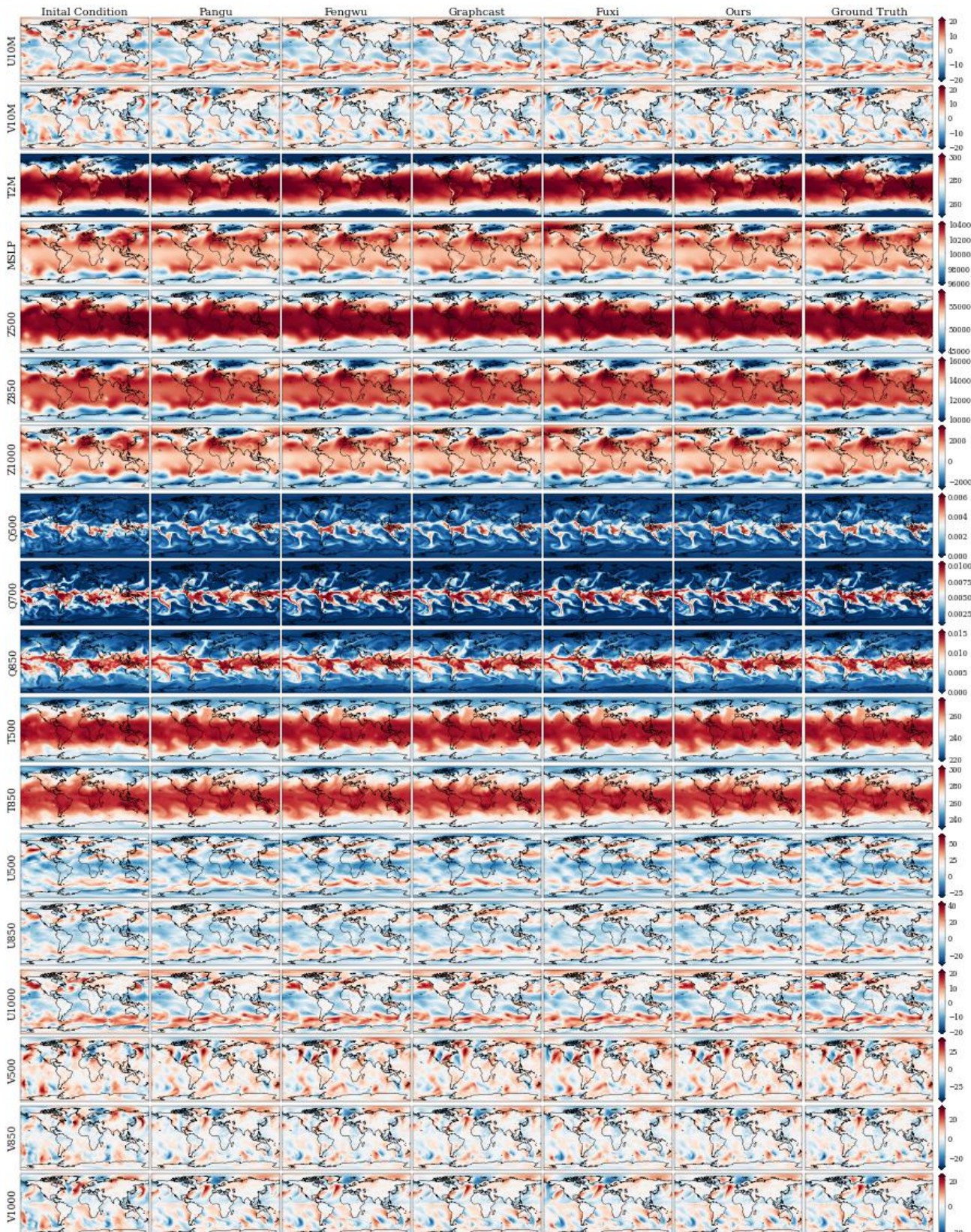

*Figure 16.* 3-day forecast results of different models.

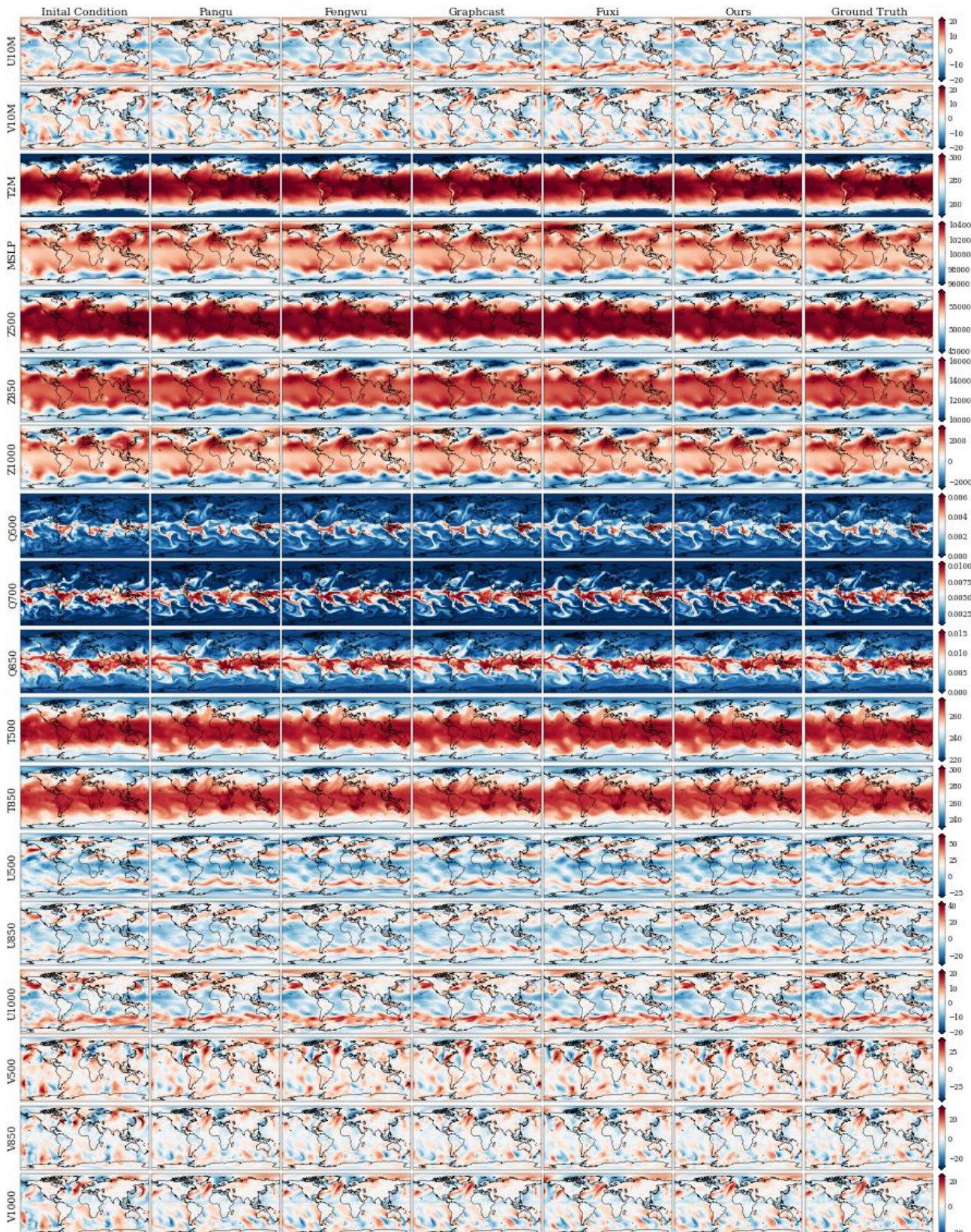

*Figure 17.* 3.5-day forecast results of different models.

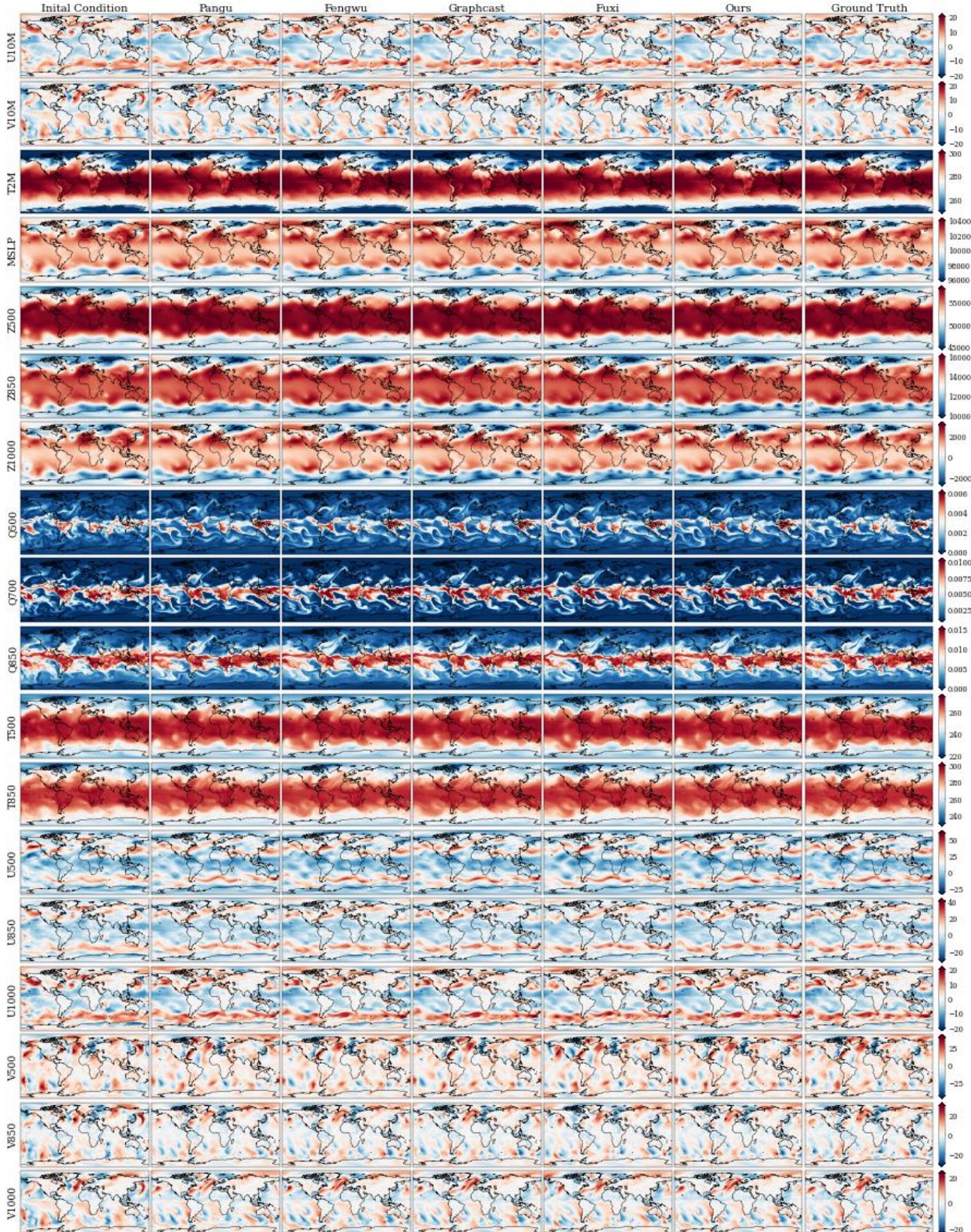

*Figure 18.* 4-day forecast results of different models.

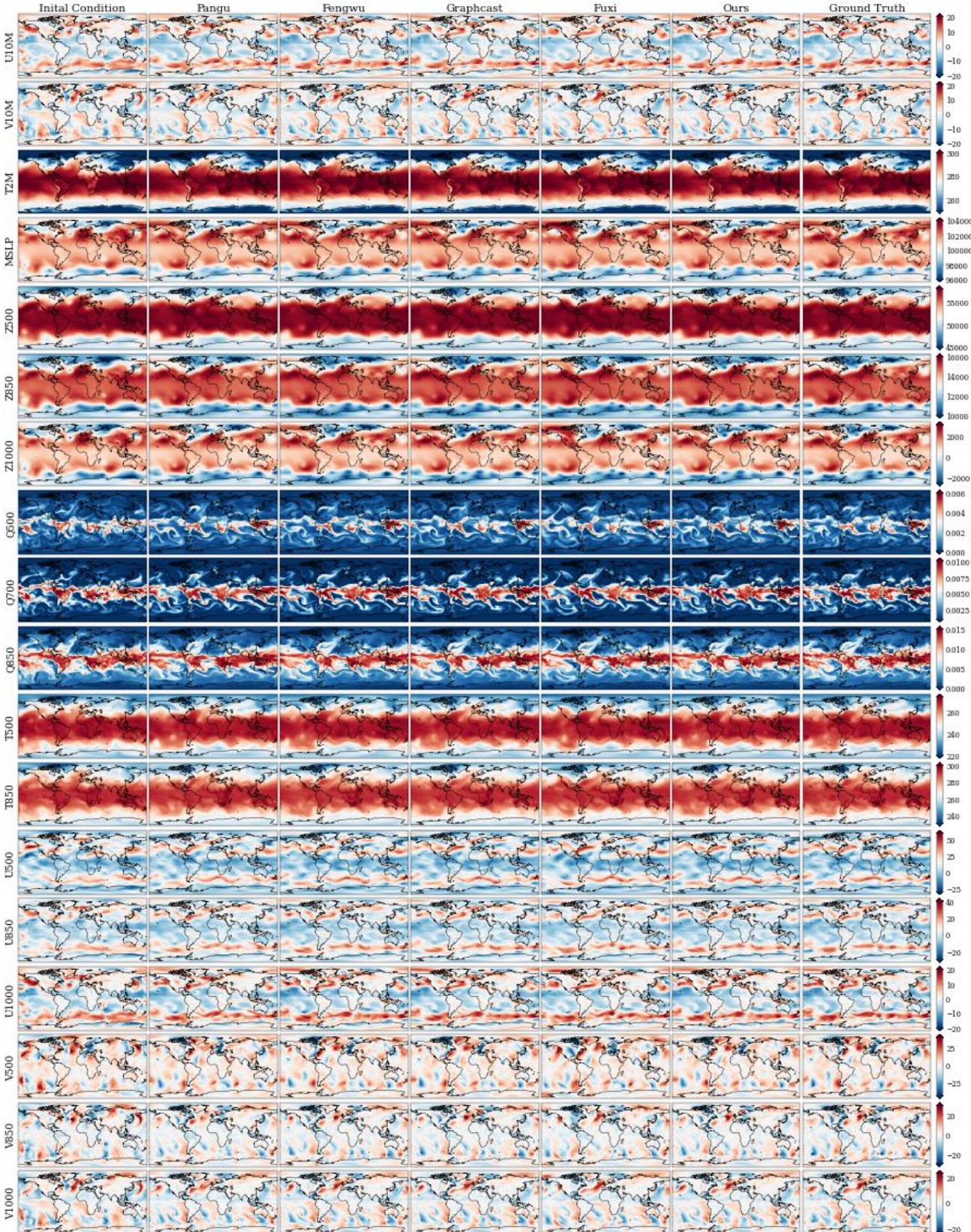

*Figure 19.* 4.5-day forecast results of different models.

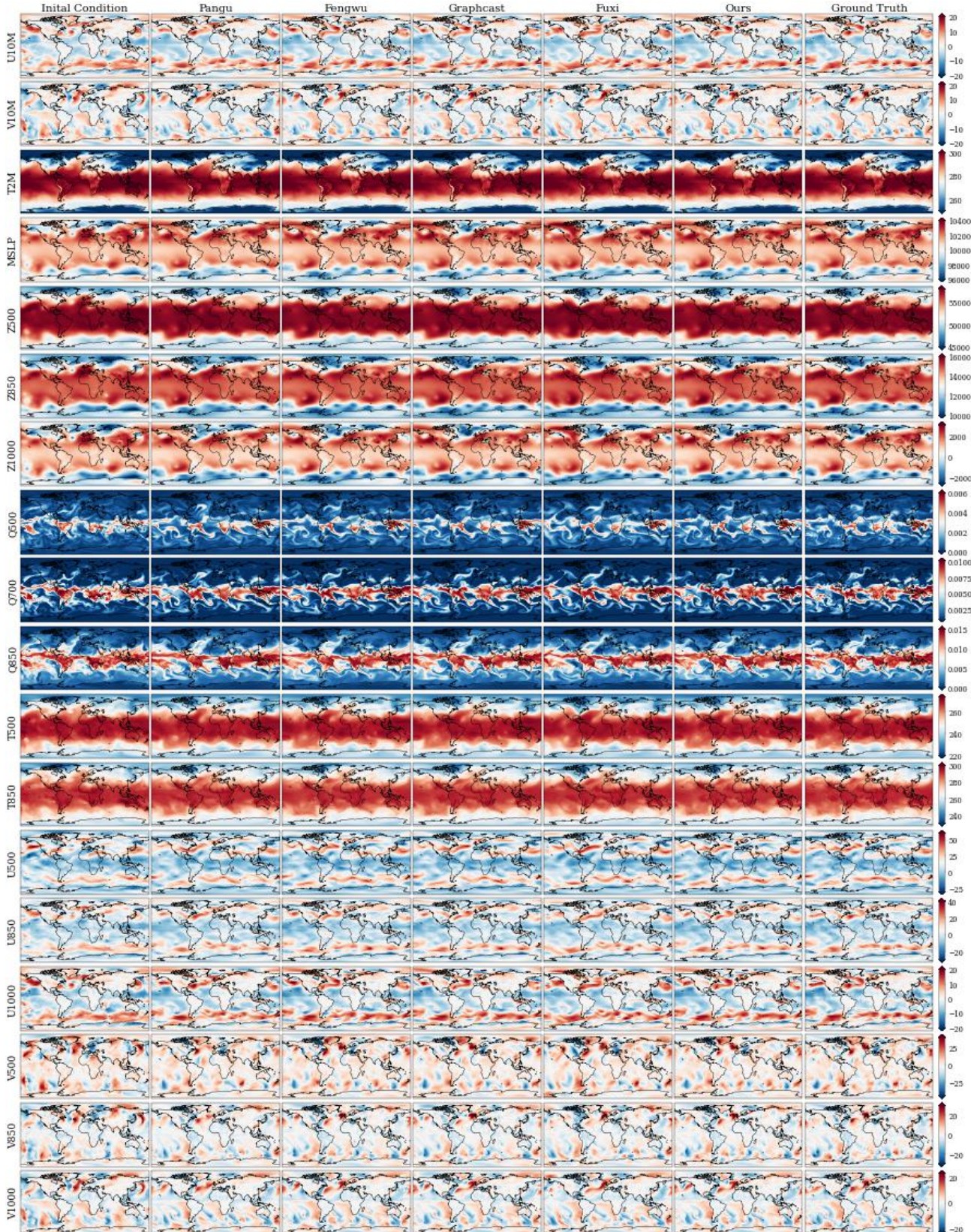

*Figure 20.* 5.0-day forecast results of different models.

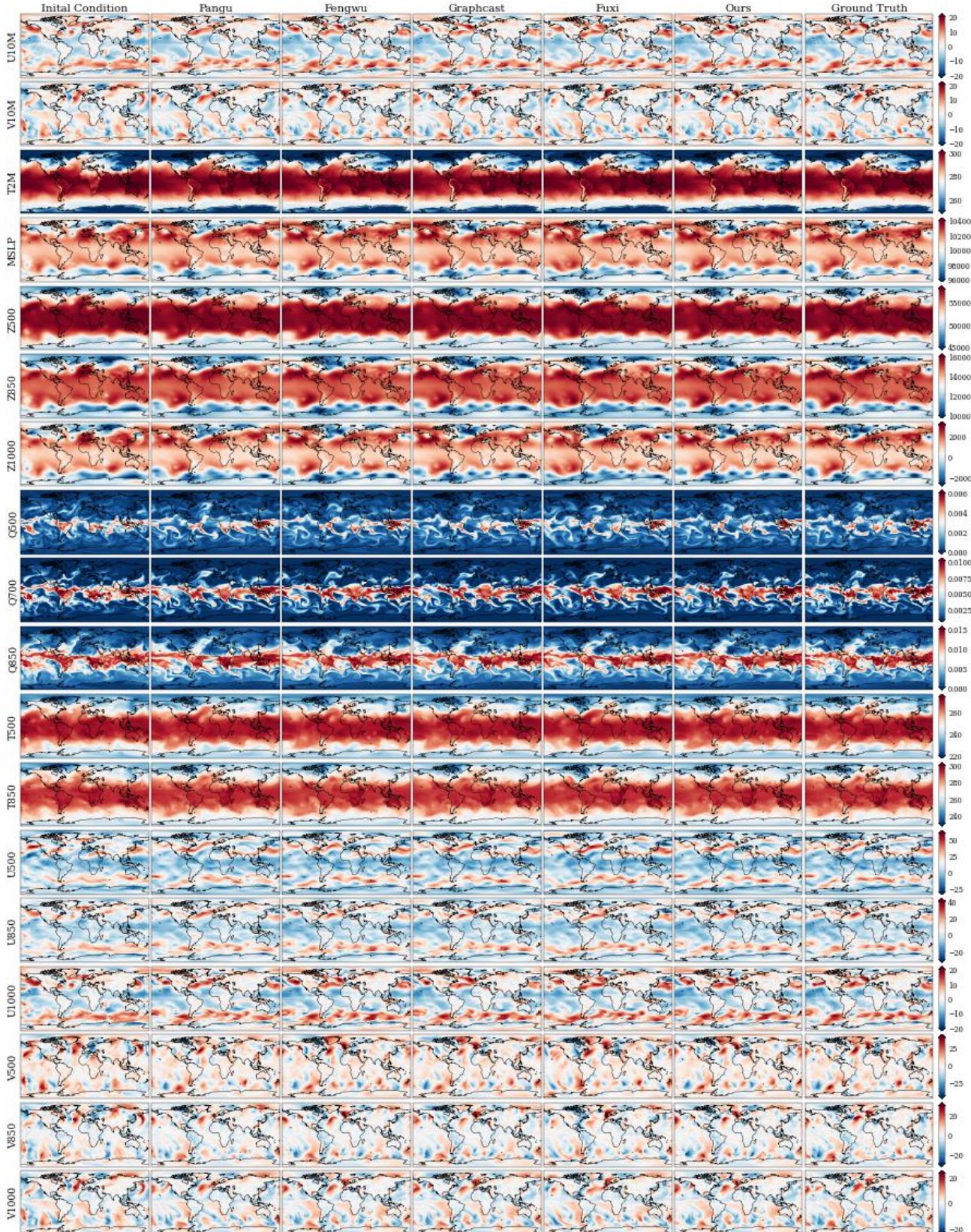

*Figure 21.* 5.5-day forecast results of different models.

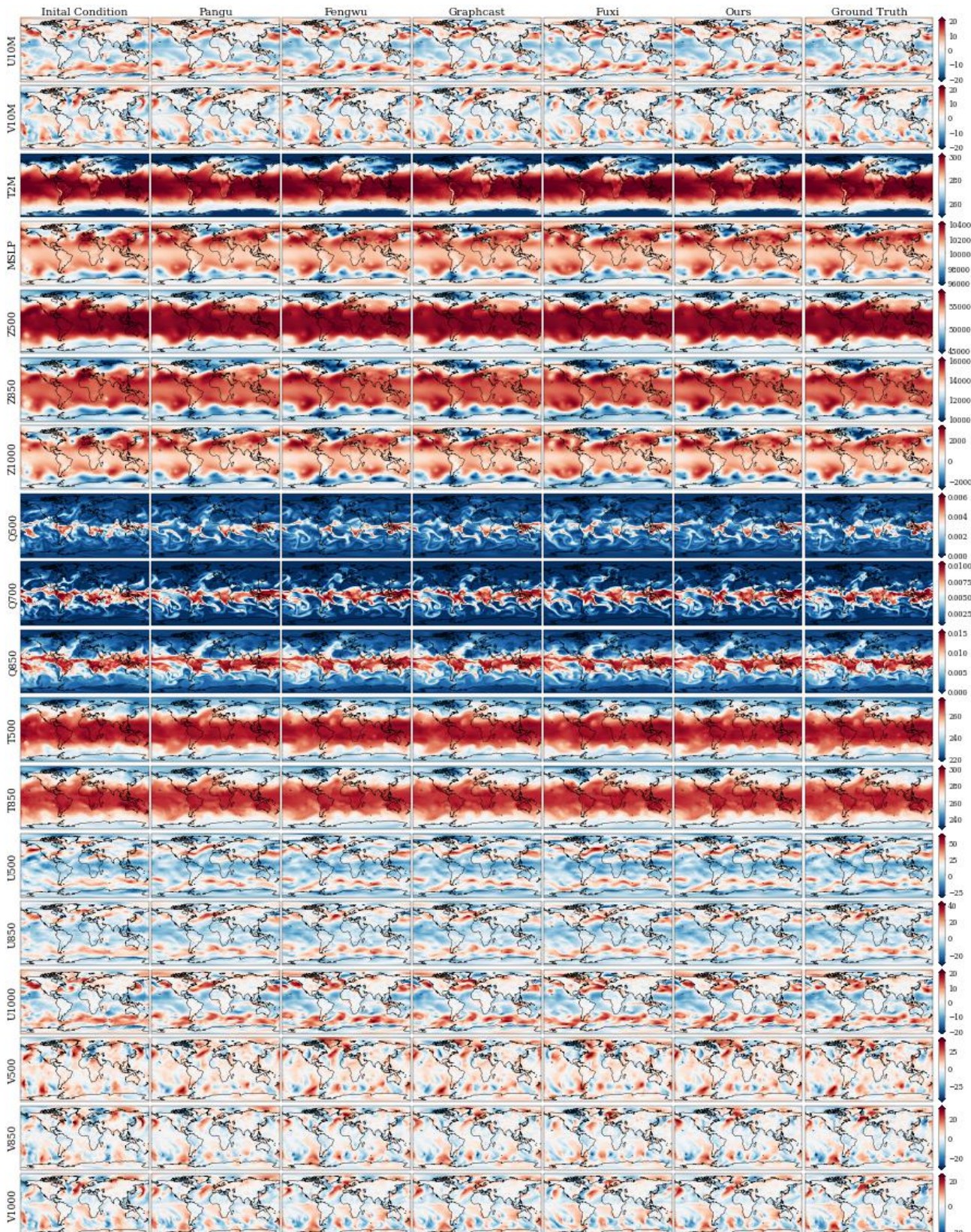

*Figure 22.* 6.0-day forecast results of different models.

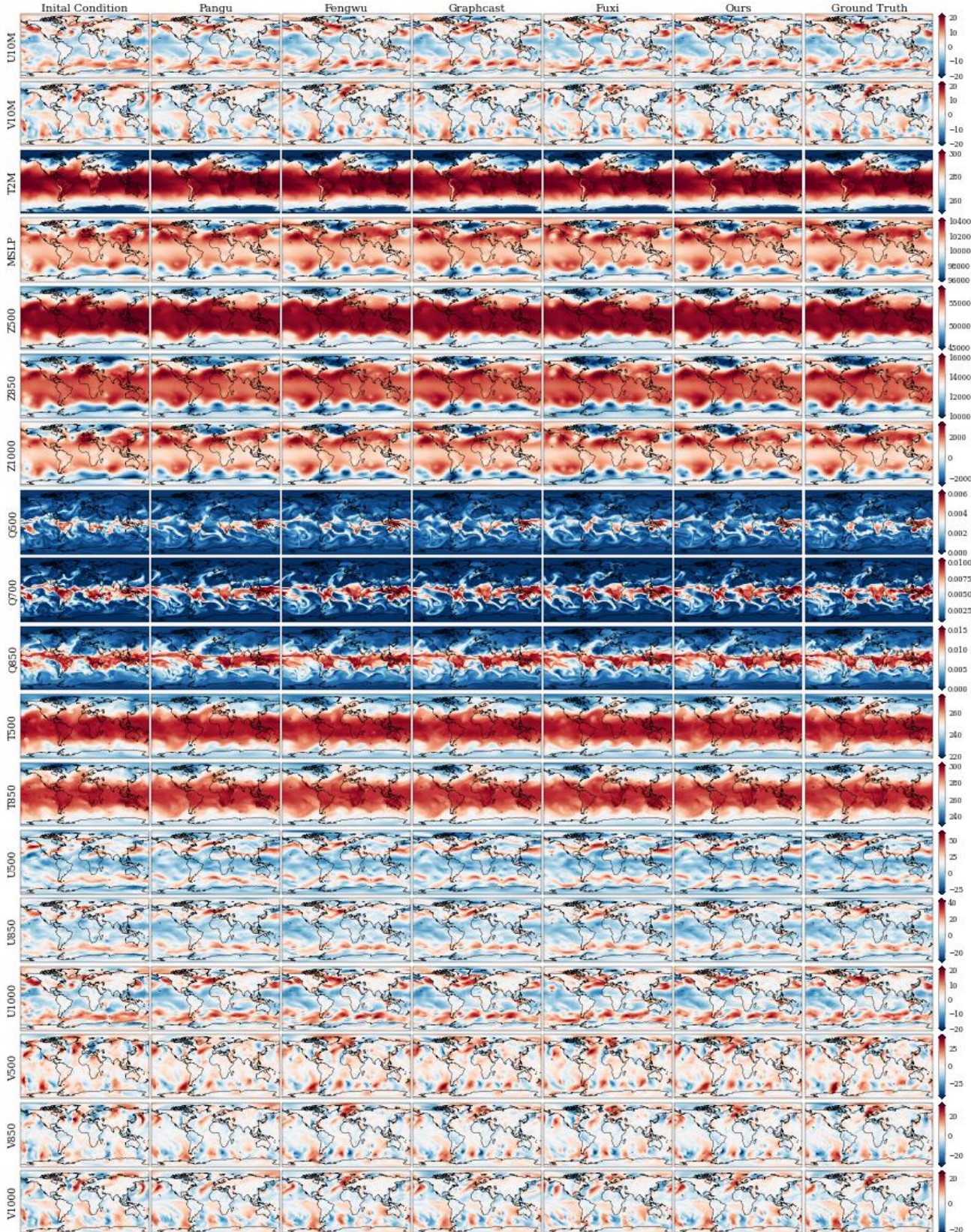

*Figure 23.* 6.5-day forecast results of different models.

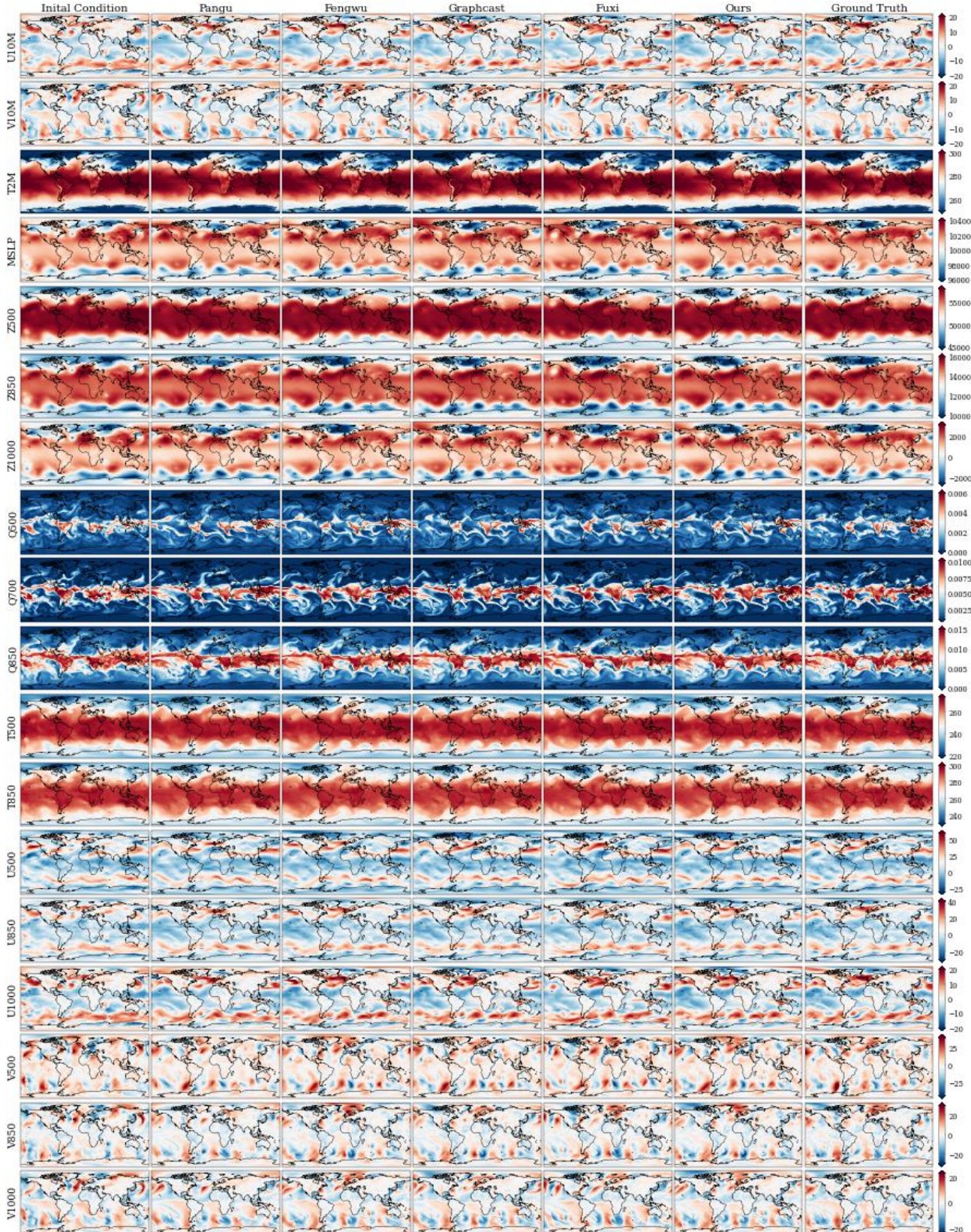

*Figure 24.* 7.0-day forecast results of different models.

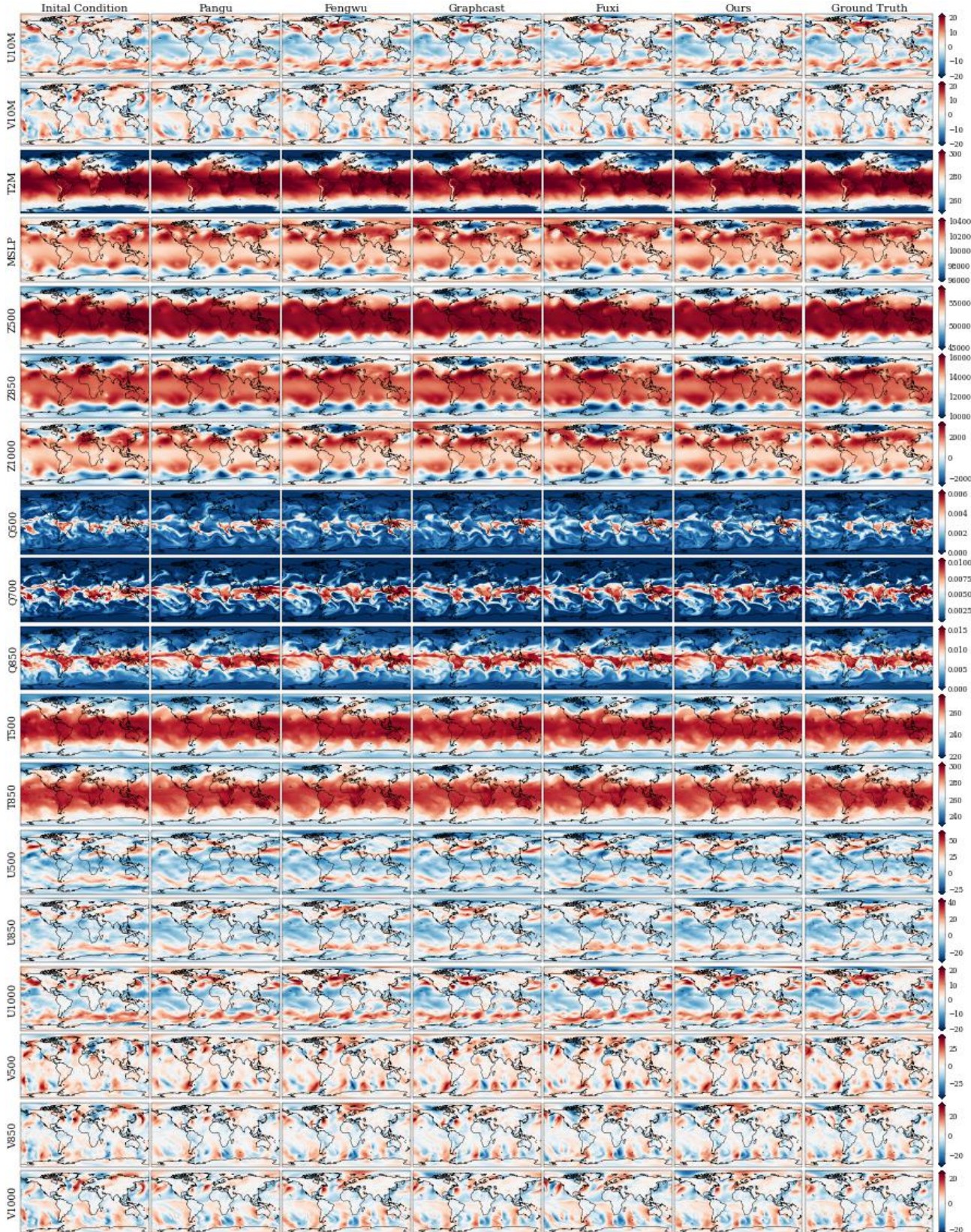

*Figure 25.* 7.5-day forecast results of different models.

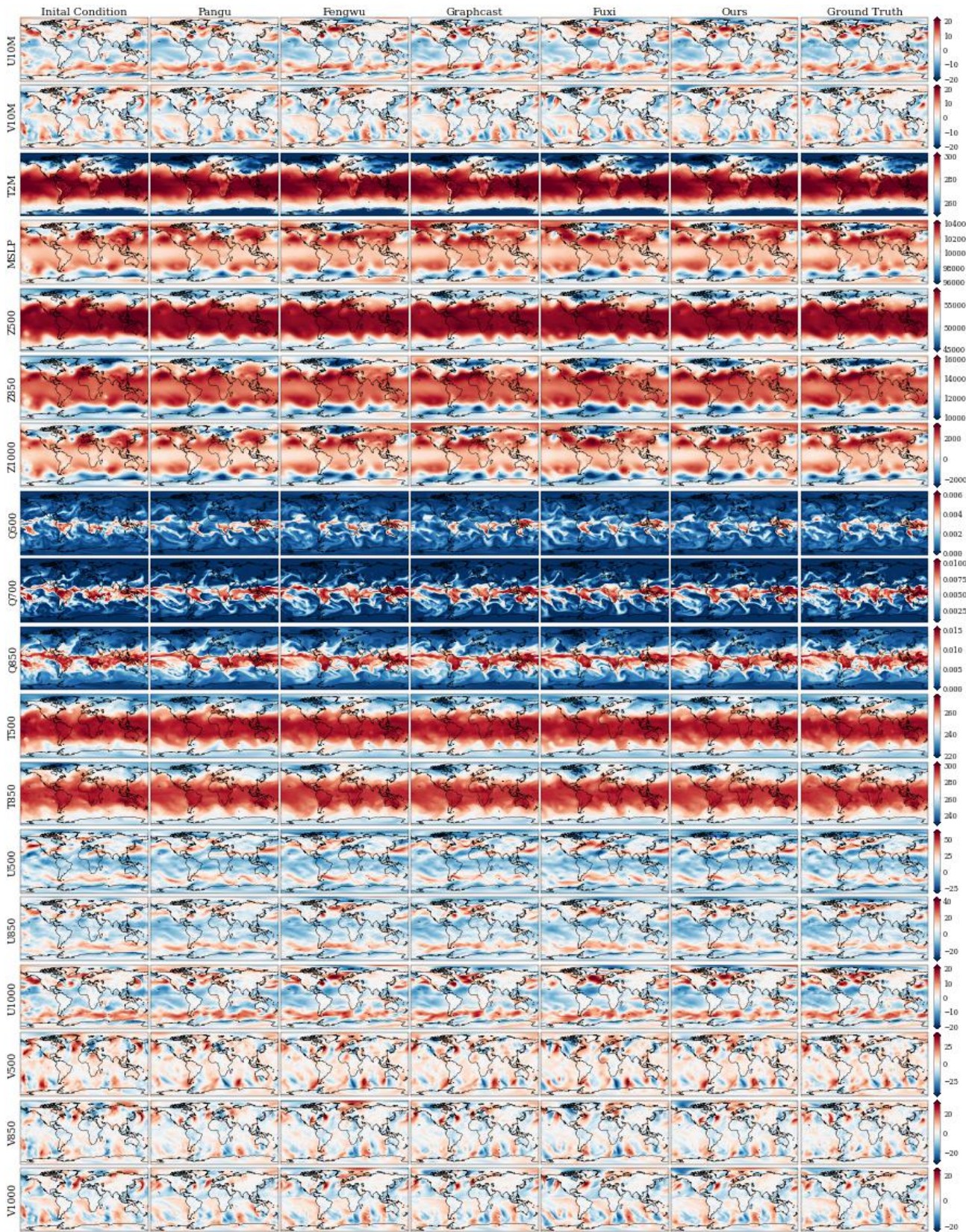

*Figure 26.* 8.0-day forecast results of different models.

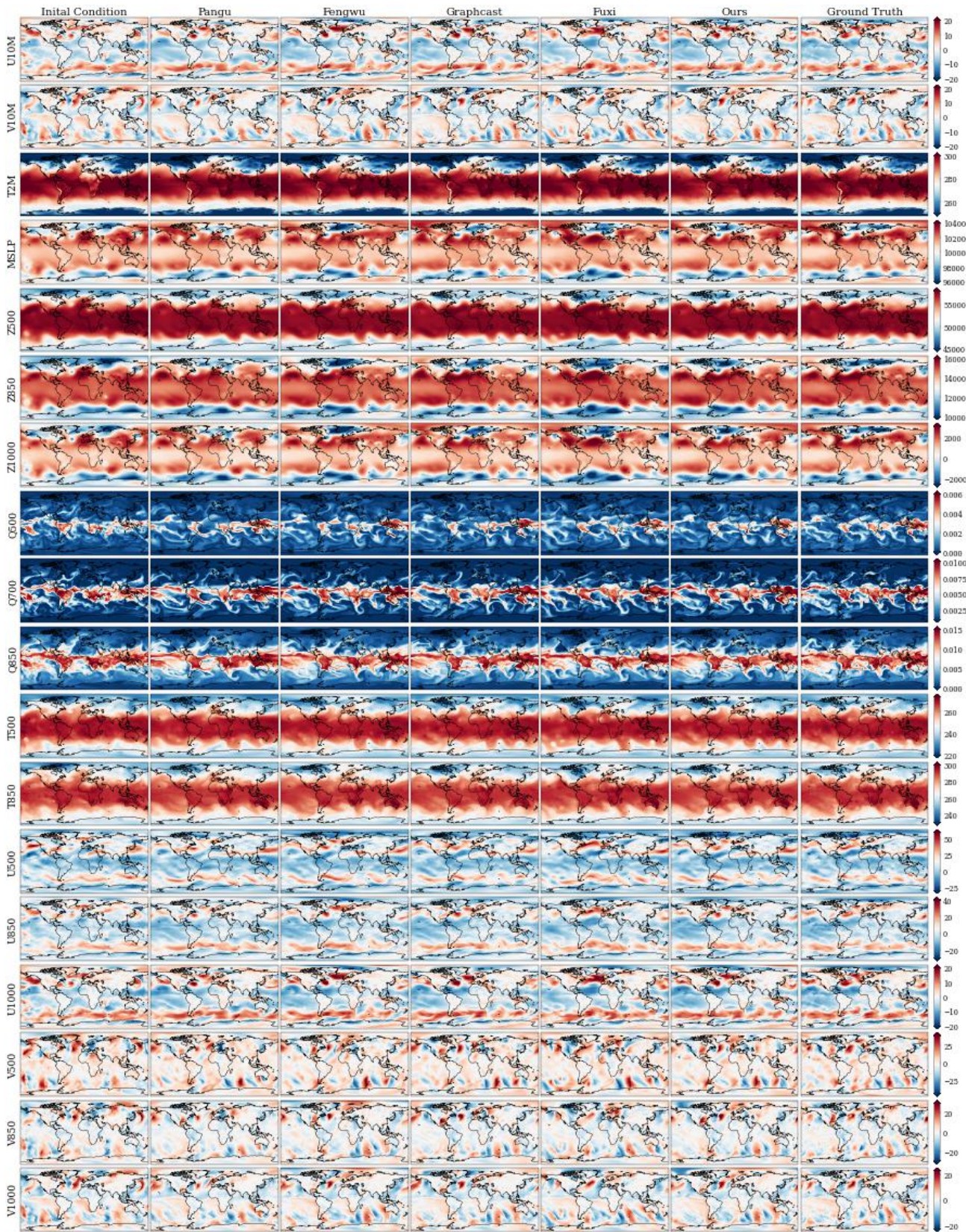

*Figure 27.* 8.5-day forecast results of different models.

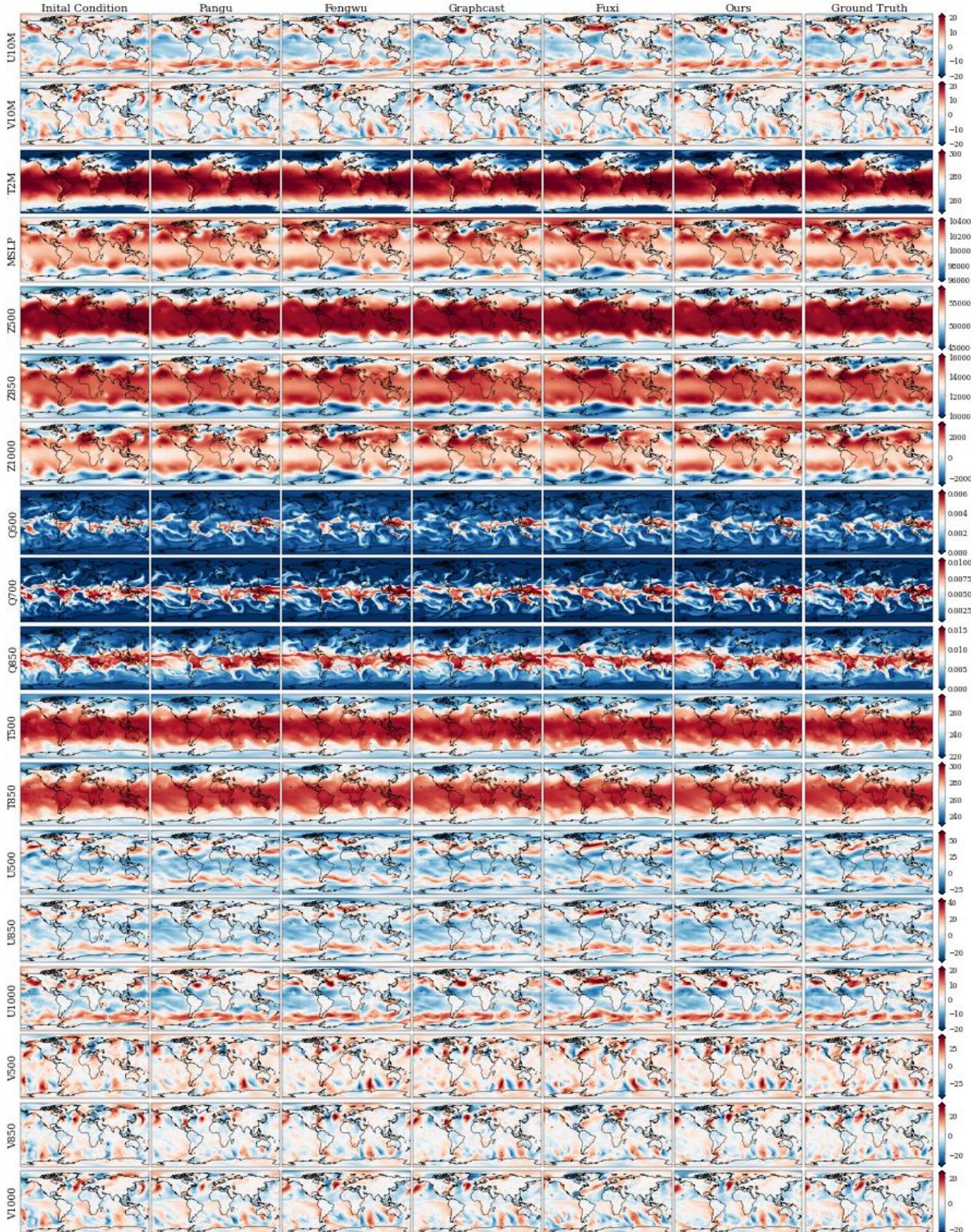

*Figure 28.* 9.0-day forecast results of different models.

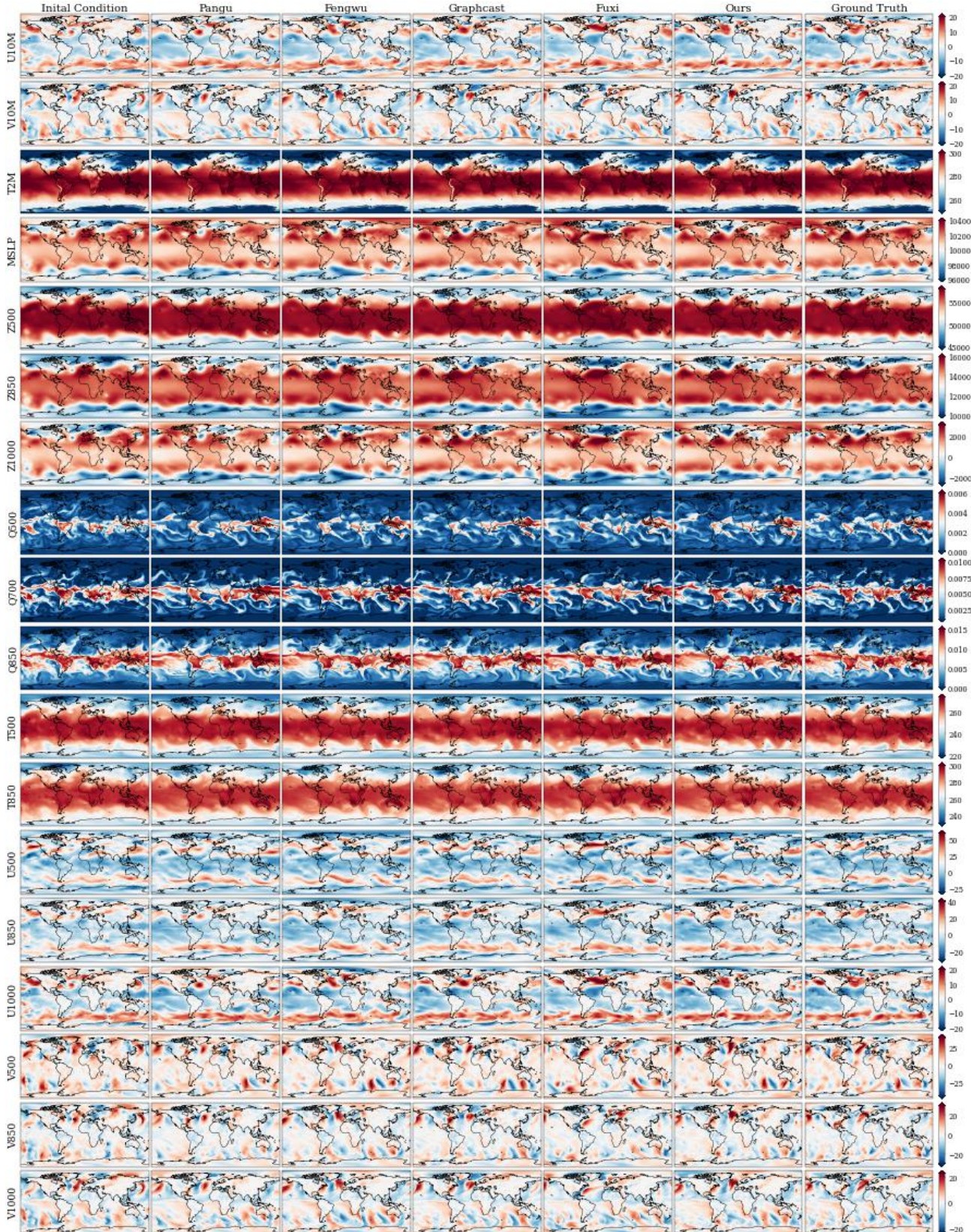

*Figure 29.* 9.5-day forecast results of different models.

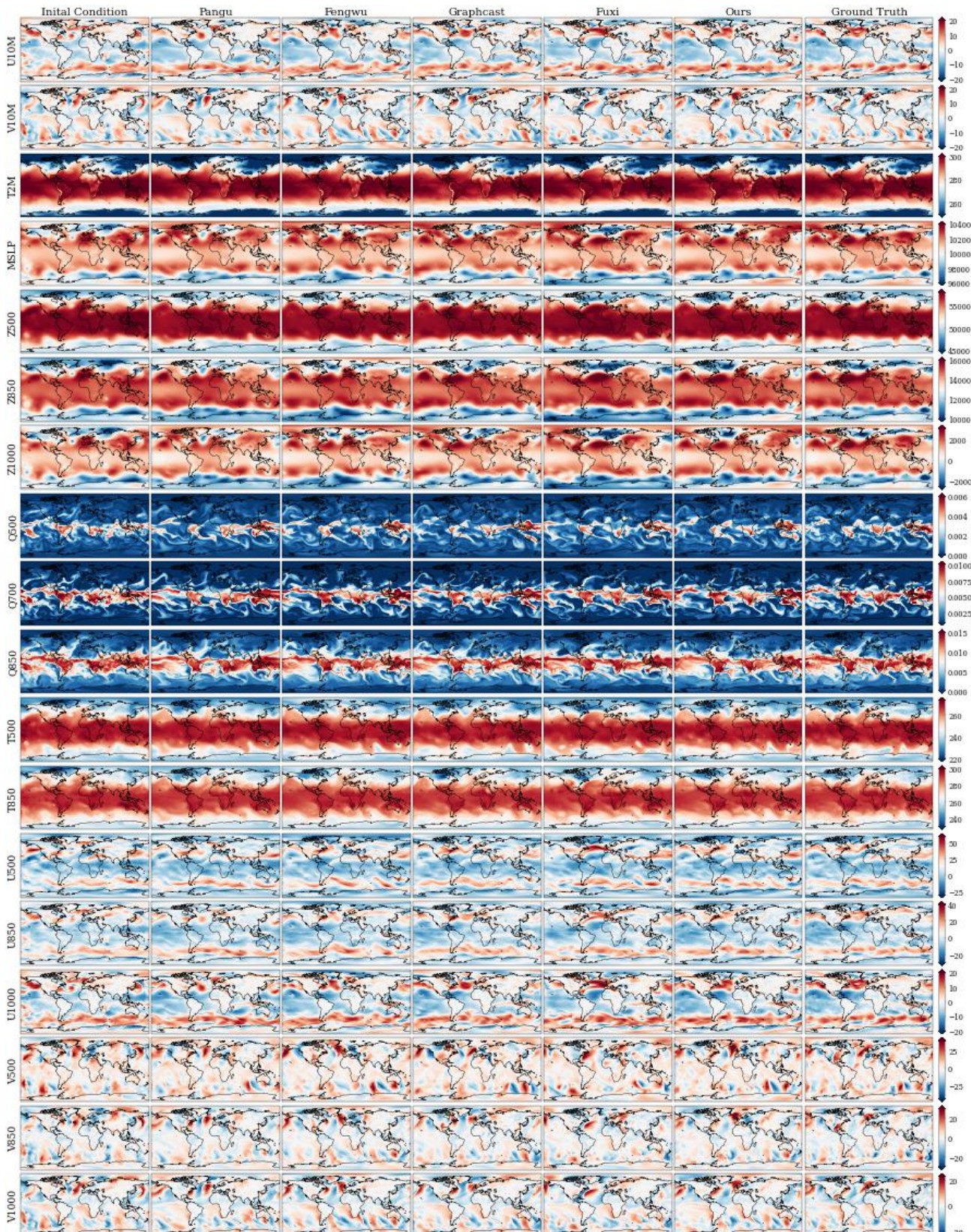

*Figure 30.* 10.0-day forecast results of different models.

