# OpenReview forum: "OneForecast: A Universal Framework for Global and Regional Weather Forecasting"
_ICML.cc/2025/Conference — ICML 2025 poster_

### Official Review · Reviewer_yiSD · 2025-02-25

**Overall Recommendation:** 3

**Summary:**

This paper introduces OneForecast which leverages multiscale graph neural networks. By integrating principles from dynamical systems with multi-grid structures, OneForecast refines target regions to better capture high-frequency features and extreme events. The adaptive information propagation mechanism, equipped with dynamic gating units, mitigates over-smoothing and enhances node-edge feature representation. Furthermore, the proposed neural nested grid method preserves global information for regional forecasts, effectively addressing the loss of boundary information and significantly improving regional forecast performance.

**Claims And Evidence:**

I partially agree with the claims made in the paper, as some of them are supported by the experiments provided. However, several claims do not seem very correct. Details will be discussed further in Questions.

**Essential References Not Discussed:**

No, the paper makes a relatively comprehensive citation of the literature.

**Experimental Designs Or Analyses:**

Yes, I have checked the work's elaboration on the setup of the experiment, the implementation method of the comparative experiment, and the analysis of other details of the experiment. Specific issues regarding the experimental section can be found in the Question.

**Methods And Evaluation Criteria:**

This work presents a relatively rich set of comparative experiments to verify the performance of OneForecast and conducts a comprehensive analysis of the experimental details. However, the evaluation criteria do not seem very correct. Details will be discussed further in Questions.

**Other Comments Or Suggestions:**

It is recommended to submit the results to WeatherBench for a standardized evaluation.

**Other Strengths And Weaknesses:**

**Strengths**
1.  This paper is well-written.
2.	The motivation of this paper is clear.
3.	The experiments are sufficient.

**Weaknesses**
1. The evaluations should be carefully improved to make them correct enough.
2. The claim for solving over-smooth challenge needs more elaboration.

**Questions For Authors:**

1.	The current model exhibits significant limitations in evaluating typhoon tracks. It is advisable to use real typhoon track data, such as best track, rather than relying on ERA5. The low resolution of ERA5 introduces substantial biases in simulating typhoons.
2.	For most evaluations in Table 1, it is difficult to directly compare each model because the results do not represent the best models available in the literature. Instead of using a normalized comparison across all variables, it would be more appropriate to evaluate each variable separately, similar to the approach used in WeatherBench. I recommend adopting the evaluation methodology employed by Pangu-Weather and GraphCast for a more meaningful comparison.
3.	The evaluation in Figure 3 has two notable issues. First, it does not include key variables of current interest. Second, the (ACC) values appear to be significantly overestimated, particularly for u10 at 10 days, which exceeds 0.7—a result that is highly impressive and potentially unrealistic. Additionally, other metrics show substantial discrepancies compared to open-source models like Pangu-Weather, raising concerns about the validity of the results.
4.	Figure 4 does not convincingly demonstrate the resolution of over-smoothing, as the results, particularly for q700, still appear excessively smooth. To address the smoothing issue more rigorously, it would be beneficial to include spectral analysis. Additionally, many of the images exhibit noticeable misalignments and artifacts, raising questions about whether the model has been sufficiently trained.
5.	Figure 6 does not sufficiently demonstrate the capability for long-term forecasts. While many models can perform long-term stable predictions, their results often converge toward a climatology. It appears that OneForecast may exhibit similar behavior, which raises questions about its ability to maintain accuracy over extended periods.
6.	For the regional prediction of extreme events in Figure 5, focusing solely on the delta of a single event is insufficient to demonstrate accurate forecasting of extreme events. Instead, it would be more informative to include statistical metrics to provide a comprehensive evaluation of the model's performance in predicting extreme events.
7.	It appears that the climatological baseline used in the ACC metrics differs from those used in other studies. This discrepancy may explain the unusual evaluation results and raises questions about the consistency and comparability of the reported metrics.

**Relation To Broader Scientific Literature:**

Some of the ideas of this paper are inspired by previous work, and it achieves better results.

**Theoretical Claims:**

From the equations and explanations provided in the paper, Oneforecast seems to be a reasonable approach.

---

> ### Author Rebuttal · Authors · 2025-04-01
>
> Dear Reviewer yiSD,
>
> We are truly grateful for the time you have taken to review our paper and your insightful review. Here we response your questions point by point.
> > Q1. The claim for solving over-smooth challenge needs more elaboration.
>
> A1. Please refer to our reply A1&A2 for Reviewer z4XB.
>
> > Q2. Submit the results to WeatherBench.
>
> A2. We will submit the results to WeatherBench after the paper is accepted.
>
> > Q3. Use real typhoon track data, such as best track, rather than relying on ERA5.
>
> A3. Please refer to our reply A2 for Reviewer SzD8.
>
> > Q4. In Table 1, whether the model has been sufficiently trained. Instead of using a normalized comparison across all variables, evaluate each variable separately, and refer to comparsion manner offered by weatherbench2.
>
> A4.
> - **The reason we retrain all models in the same framework**. Although previous ML-based models have tremendous breakthroughs, they use different experiment settings. These models have differences in terms of the spatial resolution ($1.5^\circ$, $0.25^\circ$), temporal resolution (1h, 3h, 6h, 24h), and optimization strategy (1-step training or multi-step finetune). As studied in [1], these factors seriously influence the results. However, due to the huge computing resource consumption, few works retrain them in the same framework. To fairly compare different models, in Table 1, we initially report the results of different models retrained using the same settings and use simple 1-step training for 110 epochs. We acknowledge that other tricks are beneficial to the model's performance, but our comparison is fair for all models. To rule out the models not fully converged, we will report the results of fully training 200 epochs for all models. And following your suggestion, we will also add the results released by WeatherBench2.
> - **The reason we compute normalized metircs**. To display more variables's results in the limited page, we calculate the mean of all variables. In the revision, we will show more variables individually.
>
> Following your suggestion, we show 2 types comparison, the first type is the comparison between 1-step supervised models, which includes our retrained models (Pangu, Graphcast, Fuxi, and Ours), numicial method IFS-HRES, and the model provided by Fengwu's author. Note that the input of Fengwu is 2 consecutive state, other ML-based model is 1. And the resolution of Fengwu (128×256) is higher than others (120×240). The second type is the comparison between the results released by WeatherBench2 (with many finetune tricks), which includes IFS-HRES, 2 sota models (Pangu published on Nature and Graphcast published on Science), and our model finetuned only for 1 epoch with simple multi-step supervision due to the limination of time. Note that comparisons are fair only between results of the same type. It is worth mentioning that Fengwu and our 1-step models also achieve better results than IFS-HRES. And more results are available at https://anonymous.4open.science/r/rebuttal-8C5E/RMSE_ACC.jpg
>
> #RMSE
> |Model|1-day|10-day|1-day|10-day|1-day|10-day|1-day|10-day|1-day|10-day|1-day|10-day|
> |---|:-:|:-:|:-:|:-:|:-:|:-:|:-:|:-:|:-:|:-:|:-:|:-:|
> ||U10M|U10M|V10M|V10M|U850|U850|Z200|Z200|V500|V500|V850|V850|
> |IFS-HRES|1.31|4.57|1.43|4.81|1.83|6.56|111.92|990.04|2.16|9.62|1.86|6.61|
> |Pangu|1.28|4.46|1.34|4.69|1.57|6.52|92.90|1136.54|1.98|9.67|1.60|6.49|
> |Graphcast|0.81|4.43|0.84|2.15|1.25|6.42|63.48|1028.25|1.63|9.52|1.27|6.44|
> |Fuxi|1.09|4.98|1.16|5.24|1.68|7.27|114.63|1288.08|2.28|10.92|1.73|7.20|
> |Fengwu_official|0.97|4.37|1.09|4.62|1.45|6.32|101.08|985.49|1.82|9.36|1.48|6.37|
> |Pangu_wb|1.02|4.32|1.15|4.56|1.49|6.24|103.73|948.24|1.88|9.19|1.53|6.26|
> |Graphcast_wb|0.97|4.03|1.10|4.25|1.41|5.80|102.55|888.04|1.75|8.53|1.45|5.82|
> |Ours|0.76|4.39|0.79|4.64|1.17|6.36|59.20|1003.04|1.53|9.42|1.19|6.39|
> |Ours_finetune|0.78|3.60|0.82|3.78|1.19|5.21|67.14|838.94|1.52|7.64|1.21|5.18|
>
> For the ACC results, please refer to our reply A1 for Reviewer McMn.
>
> [1] Exploring the design space of deep-learning-based weather forecasting systems.
>
> > Q5. Fig 3 does not include key variables. The evaluation manner should same as previous works.
>
> A5. For more results, please refer to Appendix Fig 8-28. The evaluation manner please refer to our reply A4.
>
> > Q6. In Fig 4, q700 is over-smoothing, please add spectral analysis.
>
> A6. Please refer to our reply A1 for Reviewer McMn.
>
> > Q7. Long-term forecast accuracy of OneForecast.
>
> A7. Please refer to our reply A4 for Reviewer McMn.
>
> > Q8. Include statistical metrics to evaluate the model's performance in predicting more extreme events.
>
> A8. For statistical metrics of typhoon and more extreme event analysis, please refer to our reply A2 for Reviewer SzD8.

---

> > ### Comment · Reviewer_yiSD · 2025-04-05
> >
> > I appreciate the author's rebuttal, which answered many of my questions, but I still have some issues:
> >
> > The performance of Fuxi and GraphCast seems different from WeatherBench. Please carefully compare all the methods with the performance from WeatherBench on both ACC and RMSE.
> > The spectral analysis in the rebuttal files seem also different from WeatherBench.
> > For now, I tend to raise the score to 2.5, which means "but could also be accepted" part of the Overall Recommendation:. If the authors address my above issues and modify this paper accordingly, I think this paper will meet the acceptance criteria of the ICML conference.

---

> > > ### Author Response · Authors · 2025-04-06
> > >
> > > > Q9. The performance of Fuxi and GraphCast seems different from WeatherBench. Please carefully compare all the methods with the performance from WeatherBench on both ACC and RMSE.
> > >
> > > A9. Thanks for your recognition of our rebuttal. Your insightful suggestion will help us improve the quality of our paper. And we want to restate that our reply A4 doesn't show the results of Fuxi from Weatherbench2. You suggested us to compare with Pangu and Graphcast released by Weatherbench2. In our reply A4, dashed lines should be contrasted with dashed lines (the first type comparison), and solid lines should be contrasted with solid lines (the second type comparison). You can check the performance of Fuxi in the first comparison type. To allay your concerns, for the second type comparison (compare with Weatherbench2), we compared all methods released by Weatherbench2 as follows, except for ENS (ensemble forecasting, not the same task) and Spherical CNN (too few ICs, only 178 compared with our used 700). So the baseline includes IFS-HRES (the best numerical method), Keisler (arXiv), Pangu (published on Nature), Graphcast (published on Science), Fuxi (published on npj Climate and Atmospheric Science), and NeuralGCM (published on Nature). Same as our reply A4, we show the average results for the first 700 ICs ('nan' means weatherbench2 doesn't release corresponding results):
> > >
> > > #RMSE
> > > |Model|1-day|10-day|1-day|10-day|1-day|10-day|1-day|10-day|1-day|10-day|1-day|10-day|
> > > |---|:-:|:-:|:-:|:-:|:-:|:-:|:-:|:-:|:-:|:-:|:-:|:-:|
> > > ||U10M|U10M|V10M|V10M|U850|U850|Z200|Z200|V500|V500|V850|V850|
> > > |IFS-HRES|1.31|4.57|1.43|4.81|1.83|6.56|111.92|990.04|2.16|9.62|1.86|6.61|
> > > |Keisler_wb|nan|nan|nan|nan|1.61|6.09|nan|nan|2.17|9.00|1.65|6.09|
> > > |Pangu_wb|1.02|4.32|1.15|4.56|1.49|6.24|103.73|948.24|1.88|9.19|1.53|6.26|
> > > |Graphcast_wb|0.97|4.03|1.10|4.25|1.41|5.80|102.55|888.04|1.75|8.53|1.45|5.82|
> > > |Fuxi_wb|0.97|3.42|1.09|3.60|1.43|4.98|nan|nan|1.78|7.30|1.46|4.99|
> > > |NeuralGCM_wb|nan|nan|nan|nan|1.53|6.13|123.71|913.29|1.69|9.06|1.40|6.17|
> > > |Ours_finetune|0.78|3.60|0.82|3.78|1.19|5.21|67.14|838.94|1.52|7.64|1.21|5.18|
> > > |Ours_finetune2|0.79|3.52|0.82|3.69|1.20|5.12|60.21|809.77|1.53|7.48|1.22|5.08|
> > >
> > > #ACC
> > > |Model|1-day|10-day|1-day|10-day|1-day|10-day|1-day|10-day|1-day|10-day|1-day|10-day|
> > > |---|:-:|:-:|:-:|:-:|:-:|:-:|:-:|:-:|:-:|:-:|:-:|:-:|
> > > ||U10M|U10M|V10M|V10M|U850|U850|Q700|Q700|V500|V500|V850|V850|
> > > |IFS-HRES|0.94|0.31|0.94|0.27|0.95|0.34|0.86|0.28|0.97|0.31|0.94|0.27|
> > > |Keisler_wb|nan|nan|nan|nan|0.96|0.36|0.91|0.35|0.97|0.33|0.95|0.29|
> > > |Pangu_wb|0.96|0.35|0.96|0.31|0.97|0.38|0.92|0.35|0.97|0.36|0.96|0.32|
> > > |Graphcast_wb|0.97|0.39|0.96|0.35|0.97|0.42|0.93|0.41|0.98|0.39|0.96|0.35|
> > > |Fuxi_wb|0.97|0.48|0.96|0.44|0.97|0.50|nan|nan|0.98|0.48|0.96|0.44|
> > > |NeuralGCM_wb|nan|nan|nan|nan|0.96|0.40|0.93|0.39|0.98|0.37|0.97|0.33|
> > > |Ours_finetune|0.98|0.42|0.98|0.38|0.98|0.45|0.94|0.38|0.98|0.42|0.98|0.38|
> > > |Ours_finetune2|0.98|0.43|0.98|0.39|0.98|0.46|0.94|0.38|0.98|0.44|0.98|0.40|
> > >
> > > More results about RMSE and ACC (1 to 10 days forecasts) can be found in this link: https://anonymous.4open.science/r/rebuttal-8C5E/RMSE_ACC_vs_weatherbench2.jpg Our primary objective is to introduce a novel paradigm for global and regional weather forecasting rather than solely optimizing metrics. While the WeatherBench2 baseline leverages numerous training strategies, we only conducted 1-epoch of finetuning ('Ours_finetune') during the brief rebuttal period. Nevertheless, a 2-epoch finetune model ('Ours_finetune2') demonstrates improved results, indicating the potential for further gains with additional finetuning. If we finetuned for more epochs, OneForecast can achieve better result. But the rebuttal time is limited, we can't finetune more epochs. Thanks for your understanding.
> > >
> > >
> > > > Q10. The spectral analysis in the rebuttal files seem also different from WeatherBench.
> > >
> > > A10. The spectral analysis of Pangu, Graphcast, and Fuxi presented in the initial rebuttal was derived from retrained models with different horizontal resolutions (0.25° vs 1.5°) compared to Weatherbench2, potentially introducing discrepancies. To ensure a fair comparison, we recomputed the surface kinetic energy spectrum and Q700 spectrum for baseline models using Weatherbench2's official results (averaged across the first 700 ICs). Our OneForecast model also achieves comparable performance in this standardized evaluation framework. Notably, as Q700 data for Fuxi were not available in the Weatherbench2, only its surface kinetic energy spectrum could be analyzed. The complete spectral analysis results are presented in this link: https://anonymous.4open.science/r/rebuttal-8C5E/spectral_analysis_vs_weatherbench2.jpg
> > >
> > > Thanks again for your insightful suggestion. We are still conducting more experiments to promote the fairness and rationality of the experiment, and we will provide a comprehensive comparison in the final version (once accepted). We kindly ask you to reconsider your rating!

---

### Official Review · Reviewer_McMn · 2025-03-10

**Overall Recommendation:** 4

**Summary:**

Accurate weather forecasting is critical for disaster preparedness and resource management, yet traditional numerical methods are computationally intensive, and deep learning approaches often struggle with multi-scale predictions and extreme events. This paper introduces **OneForecast**, a graph neural network (GNN)-based framework designed to unify global and regional weather forecasting while addressing key challenges like over-smoothing and boundary information loss.

**Claims And Evidence:**

Yes

**Essential References Not Discussed:**

I think the discussion is quite comprehensive.

**Experimental Designs Or Analyses:**

There might be an issue with the ACC-Q700 scale in Figure 4, but I'm not sure. Is it a problem of scale line offset or decimal point precision retention when drawing?

**Methods And Evaluation Criteria:**

I think this method is highly significant for the field of weather forecasting. I appreciate that the authors used a lot of computational resources to replicate all the experiments.

**Other Comments Or Suggestions:**

See weakness

**Other Strengths And Weaknesses:**

## Strengths：

1. The authors propose a global-regional nested graph neural network architecture, which is the first to implement multi-scale (from global low resolution to regional high resolution) and multi-time spans (from short-term warning to long-term forecasting) weather modeling within a unified framework.
2. The paper is well-written, with thorough experiments and theory, and the figures and tables are beautifully designed.
3. The authors have done a comprehensive replication, which is beneficial for contributing to the open-source community.

## Weaknesses

1. I'm curious about the number of parameters and efficiency. The authors should provide a comparison.
2. The temperature variable has seasonal changes, I am curious whether it will be better to subtract the climate mean from the temperature before putting it into the network. Has the author done similar experiments? Most related works did not consider this point, so there should be no problem in training without subtracting the climate mean of temperature. I am just curious whether subtracting it will improve the performance. If there is no time to make a comparison, it can be used as a future work for further exploration.

**Questions For Authors:**

See weakness

**Relation To Broader Scientific Literature:**

Sure!

**Theoretical Claims:**

The paper provides a theoretical proof, showing that the author's designed module helps enhance the model's ability to capture high-frequency information.

---

> ### Author Rebuttal · Authors · 2025-04-01
>
> Dear Reviewer McMn,
>
> We are truly grateful for the time you have taken to review our paper and your insightful review. Here we response your questions point by point.
>
> > Q1. There might be an issue with the ACC-Q700 scale in Fig 4.
>
> A1. Thank you again for your careful review of our paper, it exactly the issue with the decimal point precision retention when drawing. And we will update Fig 4 in the revision. Following the suggestion of Reviewer yiSD, we compare all our retrained models which trained for 200 epochs with the results released by weatherbench2 with the same initial conditions (the total numbers are 700). And following the suggestion of Reviewer yiSD, we add the spectral analysis of Q700 and Wind10M in this link: https://anonymous.4open.science/r/rebuttal-8C5E/spectral_analysis.jpg
>
> Below are the ACC results. More results are available at: https://anonymous.4open.science/r/rebuttal-8C5E/RMSE_ACC.jpg
>
> #ACC
> |Model|1-day|10-day|1-day|10-day|1-day|10-day|1-day|10-day|1-day|10-day|1-day|10-day|
> |---|:-:|:-:|:-:|:-:|:-:|:-:|:-:|:-:|:-:|:-:|:-:|:-:|
> ||U10M|U10M|V10M|V10M|U850|U850|Q700|Q700|V500|V500|V850|V850|
> |IFS-HRES|0.94|0.31|0.94|0.27|0.95|0.34|0.86|0.28|0.97|0.31|0.94|0.27|
> |Pangu|0.94|0.25|0.94|0.21|0.96|0.28|0.91|0.22|0.97|0.24|0.96|0.21|
> |Graphcast|0.98|0.31|0.98|0.27|0.98|0.34|0.93|0.28|0.98|0.31|0.97|0.28|
> |Fuxi|0.96|0.14|0.96|0.10|0.96|0.17|0.89|0.19|0.96|0.12|0.95|0.11|
> |Fengwu_official|0.97|0.34|0.96|0.30|0.97|0.36|0.92|0.34|0.98|0.34|0.96|0.30|
> |Pangu_wb|0.96|0.35|0.96|0.31|0.97|0.38|0.92|0.35|0.97|0.36|0.96|0.32|
> |Graphcast_wb|0.97|0.39|0.96|0.35|0.97|0.42|0.93|0.41|0.98|0.39|0.96|0.35|
> |Ours|0.98|0.33|0.98|0.29|0.98|0.36|0.94|0.30|0.98|0.33|0.98|0.29|
> |Ours_finetune|0.98|0.42|0.98|0.38|0.98|0.45|0.94|0.38|0.98|0.42|0.98|0.38|
>
> > Q2. The number of parameters and efficiency.
>
> A2. Our OneForecast has a competitive performance for Parameters and MACs. For the MACs, the size of input tensor is set to (1, 69, 120, 240). Not that for the ML-based weather forecasts, the computational cost is less important compared with the forecasting accuracy because the ML-basd model is several orders of magnitude faster (maybe tens of thousands of times) than traditional numerical methods. For instance, in numerical forecasting, a single simulation for a 10-day forecasts can take hours of computation in a supercomputer that has hundreds of nodes. In contrast, ML-based weather forecasting models just need a few seconds or minutes to produce 10-day forecasts using only 1 GPU.
>
> |Model|Params (M)|MACs (G)|
> |---|:-:|:-:|
> |Pangu|23.83|142.39  |
> |Fengwu|153.49|132.83|
> |Graphcast|28.95|1639.26|
> |Fuxi|128.79|100.96|
> |Ours|24.76|509.27|
>
> > Q3. The temperature variable has seasonal changes, I am curious whether it will be better to subtract the climate mean from the temperature before putting it into the network.
>
> A3. This question is rarely considered in previous works. And we conduct experiments to research this point. Unfortunately, we find this even leads to poorer performance of temperature forecasts. Honestly speaking, we do not know the reason, maybe the interaction between different features, just as the question proposed by Review z4XB, please refer to our reply A12 to Review z4XB. Below are the results of two training strategies for some important level temperature variables, specifically, 'Our_t' represents the training strategy that subtracts the climate mean.
>
> #RMSE
> |Model|1-day|4-day|7-day|10-day|
> |---|:-:|:-:|:-:|:-:|
> |**T500**|
> |Ours_t|0.63|1.70|2.94|3.77|
> |Ours|0.45|1.25|2.40|3.38|
> |**T850**|
> |Ours_t|0.85|1.94|3.17|4.05|
> |Ours|0.65|1.46|2.63|3.67|
> |**T2M**|
> |Ours_t|0.79|1.58|2.44|3.06|
> |Ours|0.72|1.34|2.15|2.88|
>
> #ACC
> |Model|1-day|4-day|7-day|10-day|
> |---|:-:|:-:|:-:|:-:|
> |**T500**|
> |Ours_t|0.98|0.86|0.57|0.32|
> |Ours|0.99|0.92|0.71|0.44|
> |**T850**|
> |Ours_t|0.97|0.83|0.55|0.29|
> |Ours|0.98|0.91|0.71|0.43|
> |**T2M**|
> |Ours_t|0.96|0.81|0.55|0.32|
> |Ours|0.96|0.87|0.68|0.44|
>
> > Q4. OneForecast achieves multi-time spans (from short-term warning to long-term forecasting) weather modeling within a unified framework.
>
> A4. Thanks for your agreement with our work. We want to add some details about long-term forecasts. For adequate evaluation, we conduct quantitative analysis for long-term forecast and compute the RMSE and ACC for 100-day forecast in this link: https://anonymous.4open.science/r/rebuttal-8C5E/100days_forecast.jpg
> Our model has better RMSE and ACC. Note that atmospheric prediction is limited by the chaotic nature of dynamical systems, making 100-day forecasts theoretically unattainable. This experiment mainly validates our model’s ability to preserve atmospheric physics consistency, rather than focus on numerical accuracy. Existing methods often fail in extended forecasting, with high-frequency artifacts and physical collapse. In contrast, our model maintains plausible physical fields. In general, addressing the physical collapse is the first step, the next step is to improve the accuracy.

---

> > ### Comment · Reviewer_McMn · 2025-04-03
> >
> > Thank you for your response. I also reviewed the other reviewers' observations and your rebuttals and found that most concerns have been properly addressed. I am raising my score for stronger support.

---

> > > ### Author Response · Authors · 2025-04-04
> > >
> > > Thank you for your support, your support is very important to us. If you have any other questions, please let us know!

---

### Official Review · Reviewer_z4XB · 2025-03-13

**Overall Recommendation:** 3

**Summary:**

The paper introduces OneForecast, a universal weather forecasting framework based on GNNs. It aims to improve global-regional weather forecasting by leveraging multi-scale graph structures, adaptive information propagation mechanisms, and a neural nested grid method. The proposed framework improves forecast accuracy at both global and regional levels, particularly for extreme events. Extensive experiments show that OneForecast outperforms existing state-of-the-art models such as Pangu-Weather, GraphCast, Fengwu, and Fuxi.

**Claims And Evidence:**

The challenge "Lack of dynamic system modeling capability. This is especially true for capturing complex interactions between nodes at multiple scales and learning high-frequency node-edge features." is unclear. I think Pangu and GraphCast already capture dynamic systems effectively, as evidenced by their strong performance. The paper does not explicitly define what is "dynamic system modeling capability" or why other SOTA models fail in this regard.

Similarly, the concept of "high-frequency features" is frequently mentioned but never formally defined. The paper does not discuss why these features are essential for long-term forecasting or extreme event prediction.

**Essential References Not Discussed:**

No missing references were identified.

**Experimental Designs Or Analyses:**

The long-term forecasting evaluation is too weak. The authors provide only one visualization map. A more detailed quantitative analysis is required, similar to Figure 4, including ACC and RMSE for forecasts at 10, 20, 30, ..., and 100 days. Can the model really achieve reliable and accurate 100-day forecasting?

Since the paper studies the ensemble forecasts performance, can this model also be compared with GenCast?

For Figure 1, the authors state that OneForecast is trained on 1.5$\degree$ data while other models use 0.25$\degree$ data, making the comparison potentially unfair. Can the authors provide results for other models also trained on 1.5$\degree$ data? Additionally, the typhoon tracking evaluation relies only on visualization. Is there a quantitative metric to compare different models in typhoon tracking?

**Methods And Evaluation Criteria:**

The proposed methodology is simple, straightforward, and effective. However, the evaluation is insufficient (see “Experimental Designs or Analyses”).

**Other Comments Or Suggestions:**

The regional forecasting method appears to function more as an auxiliary module rather than a core component of the framework. It lacks detailed methodological explanations, significant technical innovation, and robust experimental validation. The authors seem to even forget to write the Appendix section D Model Details for Regional Forecast to further introduce this. Given these limitations, the paper should not position OneForecast as a truly "universal" framework for both global and regional forecasting. Instead, regional forecasting should be framed as a downstream task rather than a fundamental part of the model's core design.

The paper primarily justifies its proposed modules through empirical results, but it would greatly benefit from a deeper discussion on the underlying rationale behind each module’s design.

**Other Strengths And Weaknesses:**

NA

**Questions For Authors:**

Can you further explain "And it doesn’t treats the forecasts of the global model in the region as forcing, which unable to fully utilize the information of the global model"?

What is "boundary loss" in the last paragraph of the Introduction?

The proposed approach extensively uses concatenation operations to merge different types of features. Would it be possible to conduct an ablation study to assess the individual contributions of each feature?

**Relation To Broader Scientific Literature:**

The paper tries to solve an important problem in ML-based weather forecasting and contributes to general AI for numerical simulation tasks.

**Theoretical Claims:**

I am unable to fully understand the proof of Theorem 2.1

---

> ### Author Rebuttal · Authors · 2025-04-01
>
> Dear Reviewer z4XB,
>
> We are truly grateful for the time you have taken to review our paper and your insightful review. Here we response your questions point by point.
>
> > Q1&Q2. Explicitly define dynamic system modeling capability.
>
> A1&A2: Dynamic systems modeling represents multi-scale interactions, like energy transfer between low-frequency atmospheric circulation and high-frequency vortices. Meteorological dynamics follow $\frac{dx}{dt} = f(x)$ and decompose into a spectrum $x(t)=\sum\hat{x}_ne^{i\omega_n t}$. High-frequency components ($\omega_n\gg\omega_c$) correspond to local discontinuities (e.g., vortices). Models like Pangu (Transformer-based) and GraphCast (MLP-based) act as low-pass filters[1][2][3], limiting their ability to capture high frequencies and causing errors in long-term and extreme event predictions. OneForecast’s Multi-Stream Messaging (MSM) module introduces dynamic gating to enhance high-frequency information. Gating weights $g^{(h,e)}_i,g^{(h,s)}_i,g^{(h,d)}_i\propto|\lambda_i-1|+\epsilon$ depend on spectral features, where $\lambda_i$ (graph Laplacian eigenvalues) indicates frequency. Using the frequency response function $\rho(\lambda_i)$, MSM boosts signals as $\lambda_i$ approaches 2, ensuring $\rho(\lambda_i)\geq\alpha|\lambda_i-1|$ and $\rho(\lambda_i)\geq\kappa\lambda_i$. This design improves high-frequency capture while reducing low-frequency noise, as shown in Fig 7.
>
> [1]How do vision transformers work?
>
> [2]Fourier features let networks learn high frequency functions in low dimensional domains.
>
> [3]On the spectral bias of neural networks.
>
> > Q3. Explanation of Theorem 2.1.
>
> A3. Based on graph signal spectral analysis[4], the MSM module performs adaptive high-pass filtering using dynamic gating weights ($g^{(h)}\propto|\lambda_i-1|+\epsilon$). When the graph Laplacian eigenvalue $\lambda_i\to2$, the weight amplifies high-frequency components. Conversely, when $\lambda_i\to0$, the weight suppresses low-frequency effects. The frequency response $\rho(\lambda_i)$ satisfies $\rho\geq\alpha|\lambda_i-1|$ and $\rho\geq\kappa\lambda_i$, ensuring enhanced high-frequency signals, unlike traditional GCNs[5]. This design is inspired by[6].
>
> [4]The emerging field of signal processing on graphs: Extending high-dimensional data analysis to networks and other irregular domains.
>
> [5]Semi-supervised classification with graph convolutional networks.
>
> [6]Convolutional neural networks on graphs with fast localized spectral filtering.
>
> > Q4. Quantitative analysis for long-term forecast.
>
> A4. Please refer to our reply A4 for Reviewer McMn.
>
> > Q5. Comparison with GenCast.
>
> A5. Ensemble forecasting is part of our downstream tasks. We aim to show that it can enhance accuracy. Our OneForecast isn't specifically for ensemble forecast, and we'll conduct in-depth research on it later.
>
> > Q6. For typhoon track, add 1.5° results and quantitative metric.
>
> A6. For typhoon track, high-resolution modeling enhances accuracy but increases resource demands. Oneforecast delivers superior results at lower resource costs. As suggested, we present 1.5° resolution results with quantitative analysis, detailed in our reply A2 to Reviewer SzD8.
>
> > Q7. The code lacks clear instructions.
>
> A7. We will add details after the paper is accepted.
>
> > Q8. Regional forecasts should be framed as a downstream task.
>
> A8. We have removed Appendix Section D. And in Section 2.4, we’ve defined regional model as a downstream task, that’s essentially the same as your suggestion. Extensive experiments for our regional model are shown in Fig 6, Table 3, Section 3.3, and Section 3.6. To address your concerns, we’ve incorporated your valuable feedback and revised the manuscript accordingly.
>
> > Q9. A deeper discussion of rationale design for each module.
>
> A9. Please refer to A1 for our motivation.
>
> > Q10. Explain line 094-096.
>
> A10. Our NNG method enhances Graph-EFM through global model integration. Let $A_t$ denote temporal region data, $B_t$ its boundary, $A_{t+1}$ the regional forecast, and $A^\prime_{t+1}$ the coarser global forecast (matching the spatial range of $A_{t+1}$). While Graph-EFM predicts $A_{t+1}$ using $A_t$ and $B_t$, NNG incorporates $A^\prime_{t+1}$ as dynamical forcing. Although $A^\prime_{t+1}$ lacks fine-scale details, its synoptic information improves regional forecast accuracy-unexploited in Graph-EFM. Details appear in Section 2.4 and Fig 2c (expanded in revision).
>
> > Q11. What is boundary loss?
>
> A11. This is the issue caused by overlook of boundary information, and our NNG alleviates this issue, please refer to Fig 2c, Section 2.4, and Section 3.3.
>
> > Q12. Conduct ablation study to assess the contributions of each feature.
>
> A12. GNN requires both edge/node features. We perform ablation study on feature's impacts on U10M/V10M in 4-day forecasts. Due to time constraints, we employ 5-year data (20 epochs) with averaged results from 50 ICs shown at: https://anonymous.4open.science/r/rebuttal-8C5E/ablation_study.jpg

---

> > ### Comment · Reviewer_z4XB · 2025-04-08
> >
> > Thanks for your rebuttal. Most of my questions have been addressed, and I have updated my score to weak accept.
> >
> > Regarding the 100-day forecast: I understand that “this experiment mainly validates our model’s ability to preserve atmospheric physics consistency, rather than focus on numerical accuracy.” However, physics consistency is difficult to evaluate based solely on a visualization map. Additionally, given that “atmospheric prediction is limited by the chaotic nature of dynamical systems, making 100-day forecasts theoretically unattainable,” I find the inclusion of 100-day results potentially misleading and of limited value. I suggest revising the manuscript accordingly.
> >
> > Regarding Q8. “Regional forecasts should be framed as a downstream task”: I acknowledge your point that “we’ve defined regional model as a downstream task” and that “extensive experiments for our regional model are shown.” That said, the proposed model is primarily designed for global forecasting, with only minor extensions for regional tasks. Therefore, I recommend recalibrating the claims made about the regional component, as it's only one of the downstream tasks. For example, personally I would suggest removing the regional claim from the title and instead emphasizing other core contributions.

---

> > > ### Author Response · Authors · 2025-04-08
> > >
> > > We appreciate your recognition of our work! Your constructive comments have reaffirmed the significance of our contributions. And we will revise the manuscript in accordance with your suggestions. Once the paper is accepted, **we will release all of the codes in the camera ready phase, which includes data preprocessing, training, testing, and pre-trained weights**, thereby making a modest contribution to the community.

---

### Official Review · Reviewer_SzD8 · 2025-03-14

**Overall Recommendation:** 4

**Summary:**

This paper propsoes a novel method for deep learnig based weather forecasting. The proposed method is based on graph neural networks and introduces new approaches for message passing, and for integrating high resolution and low resolution data. The proposed method outperformed exisitng method in orth short and long-term forecasts across different scales.

**Claims And Evidence:**

Yes but the paper is missing comparisons with traditional numerical methods.

**Essential References Not Discussed:**

A reference to Edge Sum MLP will be helpful to the reader

**Experimental Designs Or Analyses:**

Yes. Overall the comparisons shown in Tables and figures are convincing. Ground truth cyclone path is missing from Fig. 1

**Methods And Evaluation Criteria:**

Yes

**Other Comments Or Suggestions:**

N/A

**Other Strengths And Weaknesses:**

Paper is well written and organized. The contributions are clear and in general the results support the claims made by the authors.

Suggested edits to improve clarity:
* Fig1: add the ground truth cyclone path
* Equation 14: please be specific on what MSM means. Describe it using an algorithm or equations
* add a reference to edge sum mlp
* The term MLP appears frequently in the paper. provide more details on these networks (number of layes, activation function..)

**Questions For Authors:**

N/A

**Relation To Broader Scientific Literature:**

This paper adds new contributions to the literature in weather forecasting. The use of multi-scale integrated analysis and multi-stream messaging are novel contributions that merit further investigation and may be broadly applicable to other domains.

**Theoretical Claims:**

No

---

> ### Author Rebuttal · Authors · 2025-04-01
>
> Dear Reviewer SzD8,
>
> We are truly grateful for the time you have taken to review our paper and your insightful review. Here we response your questions point by point.
>
> > Q1. The paper is missing comparisons with traditional numerical methods.
>
> A1. We add the comparison with the traditional numerical method (IFS-HRES), which is considered the best deterministic numerical method. We choose some key variables for comparison and compute the average results of RMSE and ACC for 700 initial conditions (ICs). The first IC is 0:00 UTC Jan. 2020, and the next IC is 12 hours later. Model 'Ours' represents our 1-step supervised model, and 'Ours_finetune' represents our multi-step supervised model finetuned only for 1 epoch due to the computing resources and time constraints. We acknowledge that more advanced tricks used in previous works are beneficial to the performance, but this simple finetune strategy has proven the potential of our proposed model. Below are the RMSE and ACC results:
>
> #RMSE (the smaller the better)
> |Model|1-day|4-day|7-day|10-day|
> |---|:-:|:-:|:-:|:-:|
> |**U10M**|
> |IFS-HRES|1.31|2.29|3.63|4.57|
> |Ours|0.76|1.98|3.41|4.39|
> |Ours_finetune|0.78|1.84|2.95|3.60|
> |**V10M**|
> |IFS-HRES|1.43|2.40|3.79|4.81|
> |Ours|0.79|2.06|3.58|4.64|
> |Ours_finetune|0.82|1.91|3.09|3.78|
> |**U850**|
> |IFS-HRES|1.83|3.25|5.17|6.56|
> |Ours|1.17|2.86|4.90|6.36|
> |Ours_finetune|1.19|2.67|4.25|5.21|
> |**Z200**|
> |IFS-HRES|111.92|256.70|615.40|990.04|
> |Ours|59.20|257.09|621.66|1003.04|
> |Ours_finetune|67.14|249.26|556.53|838.94|
> |**V500**|
> |IFS-HRES|2.16|4.34|7.34|9.62|
> |Ours|1.53|3.96|7.04|9.42|
> |Ours_finetune|1.52|3.70|6.12|7.64|
> |**V850**|
> |IFS-HRES|1.86|3.28|5.22|6.61|
> |Ours|1.19|2.89|4.96|6.39|
> |Ours_finetune|1.21|2.70|4.28|5.18|
>
>
> #ACC (the higher the better)
> |Model|1-day|4-day|7-day|10-day|
> |---|:-:|:-:|:-:|:-:|
> |**U10M**|
> |IFS-HRES|0.94|0.83|0.56|0.31|
> |Ours|0.98|0.86|0.59|0.33|
> |Ours_finetune|0.98|0.88|0.65|0.42|
> |**V10M**|
> |IFS-HRES|0.94|0.82|0.54|0.27|
> |Ours|0.98|0.86|0.57|0.29|
> |Ours_finetune|0.98|0.87|0.63|0.38|
> |**U850**|
> |IFS-HRES|0.95|0.84|0.59|0.34|
> |Ours|0.98|0.87|0.62|0.36|
> |Ours_finetune|0.98|0.88|0.67|0.45|
> |**Q700**|
> |IFS-HRES|0.86|0.68|0.47|0.28|
> |Ours|0.94|0.75|0.51|0.30|
> |Ours_finetune|0.94|0.78|0.57|0.38|
> |**V500**|
> |IFS-HRES|0.97|0.86|0.60|0.31|
> |Ours|0.98|0.88|0.62|0.33|
> |Ours_finetune|0.98|0.89|0.67|0.42|
> |**V850**|
> |IFS-HRES|0.94|0.82|0.54|0.27|
> |Ours|0.98|0.86|0.57|0.29|
> |Ours_finetune|0.98|0.87|0.63|0.38|
>
> > Q2. Ground truth cyclone path is missing from Fig. 1.
>
> A2. In Fig. 1, we initially treat ERA5 as the ground truth cyclone. In the revision, we will replace the EAR5 with the ground truth produced by best track [1][2], although the result is similar. The results can be found in this link: https://anonymous.4open.science/r/rebuttal-8C5E/typhoon.jpg
>
> [1] An overview of the China Meteorological Administration tropical cyclone database.
>
> [2] Western North Pacific tropical cyclone database created by the China Meteorological Administration.
>
> And we also add a quantitative metric, a lower value represents better results:
>
> |Model|Track Position Error(km)|
> |---|:-:|
> |IFS-HRES|332|
> |Pangu 1.5°|222|
> |Graphcast 1.5°|212|
> |Pangu|231|
> |Graphcast|197|
> |Ours|157|
>
> To assess the forecast performance of more extreme events, we also show 2 extreme event assessment indicators (the higer the better) CSI and SEDI. Below are the average results for 700 ICs:
>
> |Model|Wind10M|Wind10M|T2M|T2M|
> |---|:-:|:-:|:-:|:-:|
> ||CSI|SEDI|CSI|SEDI|
> |Pangu|0.11|0.29|0.16|0.34|
> |Graphcast|0.13|0.29|0.20|0.38|
> |Fuxi|0.11|0.20|0.19|0.27|
> |Ours|0.14|0.31|0.21|0.40|
>
> > Q3. A reference to Edge Sum MLP will be helpful to the reader.
>
> A3. We will add the reference of Edge Sum MLP [3] in the revision. And in Appendix E.2 Eq 24-29, we present detailed information about Edge Sum MLP.
>
> [3] Pfaff T, Fortunato M, Sanchez-Gonzalez A, et al. Learning mesh-based simulation with graph networks.
>
> > Q4. Equation 14: please be specific on what MSM means.
>
> A4. As shown in Section 2.2, MSM means Multi-stream Messaging, which includes a dynamic multi-head gated edge update module and a multi-head node attention mechanism. And MSM is used for massaging. The corresponding equations can be found in Eq 4-14. If you have any question about MSM, please do not hesitate to let us know, we will re-write the expression of MSM according to your suggestion.
>
> > Q5. Provide details about MLP.
>
> A5. As shown in Appendix E.2 Eq 22-29, this paper uses 2 types MLP. We denote the first type as MLP(·), the number of layer is 1, the latent dim is 512, and followed by the SiLU activation function and Layernorm function. And we denote the second type as ESMLP(·), the other hyperparameters are the same as MLP(·), except ESMLP(·) transforms three features (edge features, node features of the corresponding source and destination node) individually through separate linear transformations and then sums them for each edge accordingly.

---

### Decision · Program_Chairs · 2025-05-01

**Decision:**

Accept (poster)

**Comment:**

The paper introduces a new AI weather forecasting model based on graph networks. It's main contribution is unified global and regional weather forecasting through a single model, adaptive information propagation (through specific graph modules) to capture high frequency information (this is important for things like extreme weather forecasting and long horizon forecasting)

Strengths identified:
* Important contribution - there are no unified regional-global AI weather models that are performant today. Their surrogate attempts to addresses existing challenges in weather emulation through AI models relating to high frequency information capture and long rollouts
* Significant compute was used to compare with SOTA baselines (graphcast, pangu, fengwu etc) to show their model's competitive performance
* Reviewers agreed that the paper was written well and experiments comprehensive.

Weaknesses:
* SOTA weather models are trained at 0.25 deg res - resolution is downsampled in this paper. Further, the authors do not use weatherbench2 directly for comparison for this reason. This is acceptable given their low resolution training and reviewers agreed that their framework is a notable contribution.
* Regional forecasting is also emulated with ERA5 - this is probably unrealistic. Typical datasets to be used for regional weather emulation include datasets like HRRR (see StormCast https://www.arxiv.org/abs/2408.10958 which also used boundary conditions from global models). One reviewer suggests recalibrating the regional model contribution as it lacks extensive experimental validation - the authors responded with framing it as a downstream task which is acceptable for this paper.
* Other comparisons include gencast type models (diffusion) that do not exhibit smoothing etc. However, it is acceptable that these comparisons may be too much given that cost associated with re-training diffusion models.

Several clarifications were raised in the author-reviewer discussion period. Overall, the authors rebuttal convinced reviewers to lean towards acceptance of this paper. The authors need to open-source their whole workflow which they have agreed to do.